# Fibronectin is a smart adhesive that both influences and responds to the mechanics of early spinal column development

**Emilie Guillon, Dipjyoti Das[†], Dörthe Jülich, Abdel-Rahman Hassan, Hannah Geller, Scott Holley***

Department of Molecular, Cellular and Developmental Biology, Yale University, New Haven, United States

**Abstract** An extracellular matrix of Fibronectin adheres the neural tube to the two flanking columns of paraxial mesoderm and is required for normal vertebrate development. Here, we find that the bilaterally symmetric interfaces between the zebrafish neural tube and paraxial mesoderm function as optimally engineered adhesive lap joints with rounded edges, graded Fibronectin 'adhesive' and an arced adhesive spew filet. Fibronectin is a 'smart adhesive' that remodels to the lateral edges of the neural tube-paraxial mesoderm interfaces where shear stress is highest. Fibronectin remodeling is mechanically responsive to contralateral variation morphogenesis, and Fibronectin-mediated inter-tissue adhesion is required for bilaterally symmetric morphogenesis of the paraxial mesoderm. Strikingly, however, perturbation of the Fibronectin matrix rescues the neural tube convergence defect of *cadherin 2* mutants. Therefore, Fibronectin-mediated inter-tissue adhesion dynamically coordinates bilaterally symmetric morphogenesis of the vertebrate trunk but predisposes the neural tube to convergence defects that lead to spina bifida.

**\*For correspondence:**
scott.holley@yale.edu

**Present address:** [†]Department of Biological Sciences, Indian Institute of Science Education and Research (IISER), Kolkata, India

**Competing interests:** The authors declare that no competing interests exist.

## Introduction

The vertebrate central nervous system develops from the neural tube which is created via the complex morphogenic processes of convergence and closure of the neural ectoderm during neurulation. Human neural tube defects such as spina bifida arise via a failure of the neural tube to converge and fuse along the dorsal midline. Convergent-extension cell movements along with apical constriction of neural tube cells drive neurulation at the cellular level (*Wallingford et al., 2013*; *Copp et al., 2015*). In zebrafish, the neural ectoderm converges along the dorsal midline to form a neural tube with a slit-like lumen, and during posterior trunk development studied here, neural convergence is accompanied by extension both posteriorly and along the dorsal-ventral axis but without an increase in tissue volume (*Papan and Campos-Ortega, 1994*; *Lowery and Sive, 2004*; *Harrington et al., 2010*; *Steventon et al., 2016*). At the tissue level, the underlying mesoderm is required for neural tube convergence in zebrafish (*Araya et al., 2014*). In *Xenopus*, the epidermal ectoderm is pulled toward the midline by the neural ectoderm and is necessary for neural convergence (*Davidson and Keller, 1999*; *Morita et al., 2012*). Conversely, studies of chick neurulation suggest that the epidermis pushes the neural tube closed (*Colas and Schoenwolf, 2001*). Locally varied tissue mechanics are exhibited during posterior mouse neural tube closure where a supracellular network of F-actin zippers the neural tube, creating spatially restricted zones of positive and negative strain within the tissue (*Galea et al., 2017*). Overall, these studies highlight the tissue level mechanics and inter-tissue interactions involved in neural tube morphogenesis.

Cell-cell adhesion is a crucial biomechanical process that regulates neurulation, since loss of *n-cadherin/cadherin 2* leads to neural tube convergence defects in mouse and zebrafish (*Luo et al., 2001*; *Lele et al., 2002*; *Harrington et al., 2007*). *cadherin 2* also regulates the morphogenesis and

**eLife digest** In embryos, the spinal cord starts out as a flat sheet of cells that curls up to form a closed cylinder called the neural tube. The folding tube is attached to the surrounding tissues through an extracellular matrix of proteins and sugars. Overlapping strands of a protein from the extracellular matrix called Fibronectin connect the neural tube to adjacent tissues, like a kind of biological glue. However, it remained unclear what effect this attachment had on the embryonic development of the spinal cord.

Connecting two overlapping objects with glue to form what is known as an 'adhesive lap joint' is common in fields such as woodworking and aeronautical engineering. The glue in these joints comes under shearing stress whenever the two objects it connects try to pull apart. But, thanks to work in engineering, it is possible to predict how different joints will perform under tension. Now, Guillon et al. have deployed these engineering principles to shed light on neural tube development.

Using zebrafish embryos and computational models, Guillon et al. investigated what happens when the strength of the adhesive lap joints in the developing spine changes. This revealed that Fibronectin works like a smart adhesive: rather than staying in one place like a conventional glue, it moves around. As the neural tube closes, cells remodel the Fibronectin, concentrating it on the areas under the highest stress. This seemed to both help and hinder neural tube development. On the one hand, by anchoring the tube equally to the left and right sides of the embryo, the Fibronectin glue helped the spine to develop symmetrically. On the other hand, the strength of the adhesive lap joints made it harder for the neural tube to curl up and close.

If the neural tube fails to close properly, it can lead to birth defects like spina bifida. One of the best-known causes of these birth defects in humans is a lack of a vitamin known as folic acid. Cell culture experiments suggest that this might have something to do with the mechanics of the cells during development. It may be that faulty neural tubes could close more easily if they were able to unglue themselves from the surrounding tissues. Further use of engineering principles could shed more light on this idea in the future.

segmentation of the paraxial mesoderm, which flanks the left and right sides of the neural tube and contributes to the vertebrae of the spinal column (*Radice et al., 1997*; *Luo et al., 2001*; *Horikawa et al., 1999*; *McMillen et al., 2016*; *Chal et al., 2017*). The presomitic mesoderm (PSM) is the posterior paraxial mesoderm within the extending tailbud. The PSM stiffens as it develops, and this tissue solidification requires *cadherin 2* and is important for body elongation (*McMillen and Holley, 2015*; *Zhou et al., 2009*; *Mongera et al., 2018*). *cadherin 2* is further required for ordered tailbud cell migration and mutants exhibit shortened body axes (*Lawton et al., 2013*).

An extracellular matrix of Fibronectin coats the paraxial mesoderm and mediates inter-tissue adhesion between the paraxial mesoderm and the neural tube and notochord (*Dray et al., 2013*; *Araya et al., 2016*), Fibronectin matrix assembly is dependent upon its Integrin receptors, principally Integrin α5β1 and αVβ3 (*Schwarzbauer and DeSimone, 2011*). Soluble Fibronectin is readily available to cells in the developing zebrafish trunk, but Fibronectin matrix assembly only occurs when Integrins adopt an active conformation in which they can bind Fibronectin (*Jülich et al., 2009*). Cadherin 2 represses Integrin α5β1 activation by physically associating with and stabilizing a complex of inactive Integrins on adjacent cells (*Jülich et al., 2015*; *McMillen et al., 2016*). On the surface of the paraxial mesoderm and along the somite boundaries, there is little Cadherin 2, and therefore, Integrin α5β1 is activated and a Fibronectin matrix is assembled (*Jülich et al., 2015*; *McMillen et al., 2016*). Surprisingly, loss of the Fibronectin receptors leads to a shortened body axis but does not affect cell migration (*Dray et al., 2013*; *Yang et al., 1999*). Thus, it is unclear how the Fibronectin matrix functions during body elongation and neural tube morphogenesis.

Fibronectin fibrillogenesis is an intrinsically mechanical process involving the unfolding of soluble Fibronectin dimers, non-covalent crosslinking and progressive assembly into larger fibers. Fibronectin fiber assembly is promoted by cell actomyosin contractility, and fiber orientation aligns with cell traction forces in 2D cell culture (*Zhong et al., 1998*; *Lemmon et al., 2009*). Similarly, constitutively active myosin regulatory light chain activates Integrin α5, colocalizes with the activated Integrin, and induces Fibronectin matrix assembly in the zebrafish paraxial mesoderm (*Jülich et al., 2015*;

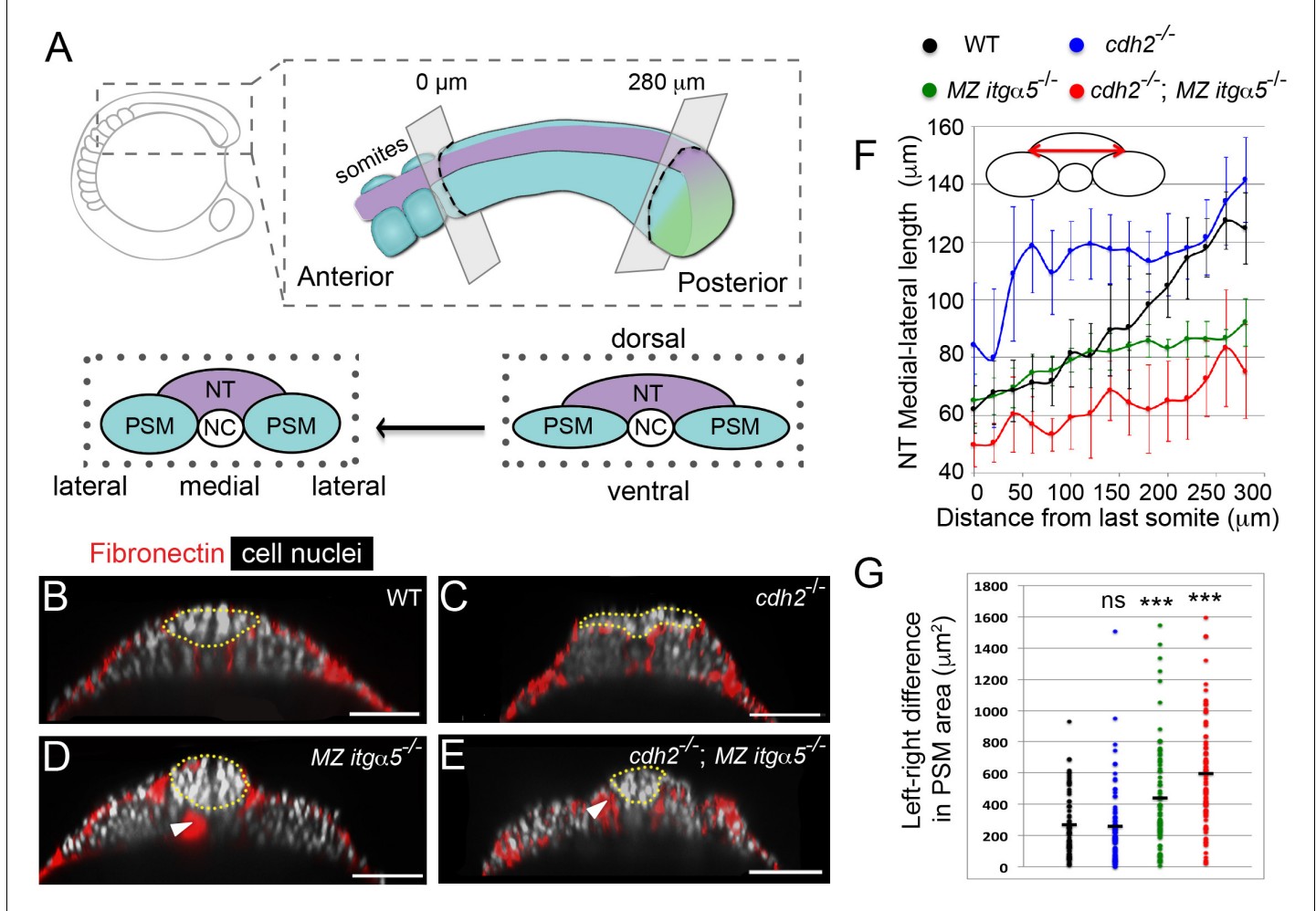

**Figure 1.** Reduction of Fibronectin matrix enhances neural tube convergence but abrogates bilaterally symmetric paraxial mesoderm morphogenesis. (A) A schematic of the zebrafish tailbud and two transverse sections at the anterior and posterior ends of the presomitic mesoderm (PSM, cyan). The left and right PSM flank the neural tube (NT, magenta) and notochord (NC). The neural tube and PSMs converge along the medial-lateral axis, and the anterior tailbud is further converged than the less developed posterior tailbud. (B–E) Transverse sections 160 µm posterior to the last somite boundary at 12–14 somite stage in wt (B), $cdh2^{-/-}$ (C), $MZ\ itg\alpha5^{-/-}$ (D), and $cdh2^{-/-}$; $MZ\ itg\alpha5^{-/-}$ (E). Sections were reconstructed at a distance of 160–180 µm from last somite boundary after wholemount labeling for fibronectin (red) and nuclei (grey). Yellow dotted lines delineate neural tube contours. White arrowheads indicate locations of tissue detachment (also see *Figure 1—figure supplement 2E and F*). Dorsal is to the top. Scale bars = 70 µm. (F) Quantification of the medial-lateral length of the neural tube (as indicated by red double arrow) along the anterior-posterior axis starting from the last somite boundary. Quantifications were performed on transverse sections spaced every 20 µm. Dots represent means and error bars represent SD. Sample size: n = 10 PSMs on five embryos for each genotype. (G) Quantification of left-right asymmetry in PSM area. Each dot denotes an absolute difference in left and right PSM areas at each transverse section. Sample size: n = 75 sections from five embryos for each genotype. ***p<0.0005, T-test. $cdh2^{-/-}$ vs WT, p=0.79; $MZ\ itg\alpha5^{-/-}$ vs WT, p=2.51e-4; $MZ\ itg\alpha5^{-/-}$;$cdh2^{-/-}$ vs WT, p=3.34e-10.

The online version of this article includes the following figure supplement(s) for figure 1:

**Figure supplement 1.** Reduction of cell-ECM interactions leads to precocious neural tube convergence and rescues *cdh2* mutant neural tube convergence defects.

**Figure supplement 2.** Reduction of cell matrix interactions provokes a precocious neural tube convergence and generates left-right asymmetries in the PSM|NT interfacial length and angle.

*McMillen et al., 2016*). In 3D in vitro micro-tissues, Fibronectin fibers and F-actin co-localize with tissue stress (*Legant et al., 2009*). These results parallel the finding that tissue tension promotes Fibronectin matrix assembly in the *Xenopus* gastrula (*Dzamba et al., 2009*).

Here, we examined the role of Fibronectin-mediated inter-tissue adhesion in neural tube convergence and paraxial mesoderm morphogenesis in zebrafish (*Figure 1A*). We first performed

morphometric analysis of the neural tube and paraxial mesoderm in different genetic backgrounds with reduced cell-cell and/or cell-Fibronectin adhesion. These experiments revealed that inter-tissue adhesion resists neural tube convergence. We developed a simple computational model that predicts several morphological changes due to loss of cell-cell or cell-Fibronectin adhesion. We find that the interfaces between the neural tube and paraxial mesoderm recapitulate features of an 'adhesive lap joint' which is commonly used in engineering and is comprised of partially overlapping components bound via an adhesive. Excess adhesive that can ooze from the edges of the overlapping domains is called an 'adhesive spew' which can be filleted or sculpted to strengthen the lap joint. Here, the Fibronectin matrix functions as the adhesive in the lap joint formed by the neural tube and left and right paraxial mesoderm. Our computational model, as well as lap joint theory, predicts different mechanical properties for the interface between the paraxial mesoderm and the neural tube compared to the interface between the paraxial mesoderm and epidermis. Morphometric analysis of the Fibronectin matrix and the actomyosin cytoskeleton are consistent with the prediction that there is an increasing medial to lateral gradient of tension along the paraxial mesoderm and neural tube interface. Lastly, we created a photoconvertible Fibronectin transgenic that enables us to paint the extracellular matrix and quantify its remodeling in live embryos. These experiments indicate the Fibronectin matrix continually remodels to the lateral interface of the neural tube and paraxial mesoderm where tension is highest. This remodeling is dependent upon inter-tissue adhesion and convergence of the neural tube, implying that there is shear stress along this tissue interface that drives the remodeling. Moreover, these experiments revealed that Fibronectin matrix remodeling on one side of the body is sensitive to morphogenic variation on the contralateral side of the body due to mechanical coupling via inter-tissue adhesion. Altogether, the data indicate that Fibronectin-mediated inter-tissue adhesion acts as an adhesive lap joint that mechanically coordinates bilaterally symmetric morphogenesis but predisposes the neural tube to convergence defects that lead to spina bifida.

## Results

### Reduction of fibronectin matrix fosters neural tube convergence but abrogates bilaterally symmetric PSM morphogenesis

We quantified neural tube (NT) morphologies in wild type, *cadherin 2* mutants (*cdh2$^{-/-}$*), maternal zygotic *integrin α5* mutants (*MZ itgα5$^{-/-}$*) and *cdh2$^{-/-}$*; *MZ itgα5$^{-/-}$* double mutants. We fixed embryos at 12–14 somite-stage and imaged them for Fibronectin and cell nuclei. *cdh2* mutants exhibited a wider neural tube than wild-type embryos consistent with its known convergence defect (*Figure 1B–C*; *Lele et al., 2002*). In contrast, *MZ itgα5* mutants exhibit a narrower neural tube (*Figure 1D*) suggesting precocious neural tube convergence after reduction of cell-Fibronectin matrix adhesion. Surprisingly, the convergence defect of *cdh2* mutants is rescued in *cdh2$^{-/-}$*; *MZ itgα5$^{-/-}$* double mutants (*Figure 1E*).

We quantified convergence of the neural tube along the anterior-posterior axis across genotypes. We measured the medial-lateral width of the neural tube (cartoon in *Figure 1F*) on transverse sections every 20 μm along the anterior-posterior axis starting from the last somite boundary until the posterior end of the PSM (*Figure 1—figure supplement 1*). In wild-type embryos, the width of the neural tube progressively narrows, comparing posterior to anterior, reflecting convergence over time (*Figure 1F* and *Video 1*). This change in the medial-lateral width of the neural tube is shallower in mutants.

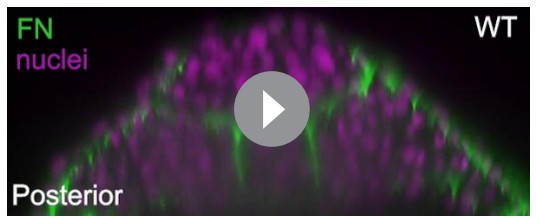

**Video 1.** Changes in neural tube and PSM shapes over developmental time. This movie is a series of transverse sections from posterior to anterior from a fixed embryo showing Fibronectin localization (green) and cell nuclei (magenta). The development of the vertebrate trunk and tail proceeds from anterior to posterior, thus this series illustrates the medial convergence and shape changes of the neural tube and PSM over time. The quantification of these shapes in different genotypes is detailed in *Figures 1*, *2* and *5*, *Figure 1—figure supplement 1*, *Figure 1—figure supplement 2*, *Figure 2—figure supplement 1* and *Figure 5—figure supplement 1*.
https://elifesciences.org/articles/48964#video1

In *cdh2* mutants, the neural tube is wider than wild-type embryos along the full anterior-posterior length of the PSM (*Figure 1F*). In *MZ itgα5* mutants and *cdh2; MZ itgα5* mutants, the neural tube has a narrower medial-lateral width in the posterior PSM compared to wild type (*Figure 1F*). Similarly, wild-type embryos exhibited a posterior to anterior decrease in the dorsal-ventral length and the cross-sectional area of the neural tube, while the posterior to anterior decreases in these two metrics were shallower in other genotypes (*Figure 1—figure supplement 2A and B*). These data suggest that inter-tissue adhesion via cell-Fibronectin matrix adhesion constrains neural tube convergence.

We examined the effect of loss of inter-tissue adhesion on the PSM and observed an abrogation of bilaterally symmetric morphogenesis. We measured the left-right differences in PSM cross-sectional areas (*Figure 1G*) and found that both *MZ itgα5$^{-/-}$* and *cdh2$^{-/-}$; MZ itgα5$^{-/-}$* double mutants exhibited loss of bilateral symmetry. By contrast, *cdh2* mutants did not have symmetry defects. Local asymmetries were also observed in *MZ itgα5$^{-/-}$* and *cdh2$^{-/-}$; MZ itgα5$^{-/-}$* double mutants in the length of the interface between PSM and neural tube (PSM|NT interface) and the angle formed at the lateral edge of this interface (*Figure 1—figure supplement 2C and D*). An explanation of these asymmetries is that reduction of cell-matrix interactions creates local tissue detachments (arrowheads in *Figure 1D and E*, *Figure 1—figure supplement 2E and F*) which in turn leads to disequilibrium in inter-tissue adhesion along the left and right PSM|NT interfaces.

## A computational model relates tissue shape changes to tissue mechanics

To understand how variable levels of cell-cell adhesion and inter-tissue adhesion underlie the differences in tissue shapes across genotypes, we designed a computational 2D model with variable cell-cell adhesion and inter-tissue adhesion and tested whether the model can predict tissue shapes. We modeled a 2D transverse section of four connected tissues (NT, left and right PSM and notochord [NC]) (see Materials and methods for details). In this coarse-grained model, we do not directly consider individual cells, rather the tissues are considered as soft units with an internal pressure (blue arrows in *Figure 2A*) and an elastic surface made of movable points interlinked by springs (green surface springs in *Figure 2A*). These connected surface springs model tissue surface tension. This tissue surface tension resists the internal pressure and was shown to be proportional to the cell-cell adhesion strength inside a tissue (*David et al., 2014*; *Manning et al., 2010*). The tissues are attached to each other by another set of springs (red springs in *Figure 2A*), which model inter-tissue adhesion via cell-Fibronectin interactions. Overall, there are three parameters in the model: the surface stiffness (spring constant $K_S$ for surface springs), the adhesion stiffness (spring constant $K_{adh}$ for adhesive springs), and the internal pressure (P). The surface stiffness and the adhesion stiffness positively correlate with the levels of Cadherin-dependent cell-cell adhesion and Fibronectin-mediated inter-tissue adhesion, respectively.

To determine how the model parameters affect tissue morphology, we analyzed the simulated tissue shapes. The internal tissue pressure should depend on the net fluid content inside a tissue, and there is no obvious reason to vary this quantity across genotypes. Hence, we fixed the internal pressure (p=5) depending on an earlier model of soft grains (*Åström and Karttunen, 2006*) in such a way that the simulated shapes qualitatively resemble in vivo tissue shapes in 2D transverse sections (*Figure 2B*). Irrespective of choices of $K_s$ and $K_{adh}$, we found that the PSM|NT and the PSM and epidermis interface (PSM|E) always have very different curvatures (*Figure 2C*). The PSM|NT interface is straight and has much higher radius of curvature than the PSM|E interface which is curved (*Figure 2C*). This prediction was confirmed experimentally in vivo (*Figure 2C*).

The parameters $K_s$ and $K_{adh}$ were systematically varied in simulations to assess their influence on the shape of the interface between NT and PSM (*Figure 2—figure supplement 1*). We measured two shape metrics: the length of the interface between the PSM and neural tube (L PSM|NT, blue boxes in *Figure 2B*) and the angle at the lateral edge of this interface (angle Θ in *Figure 2B*). We found that increasing surface stiffness decreases the interfacial length, while increasing inter-tissue adhesion increases interfacial length (*Figure 2—figure supplement 1A and B*). Thus, inter-tissue adhesion acts like a 'zipper' between two tissues. In fact, low adhesion stiffness decouples the tissues, which is evident from the local detachments seen in our simulations (*Figure 2B* top right, arrowhead) and similar to detachments observed in *MZ itgα5* mutants and *cdh2; MZ itgα5* mutants (*Figure 1D and E*).

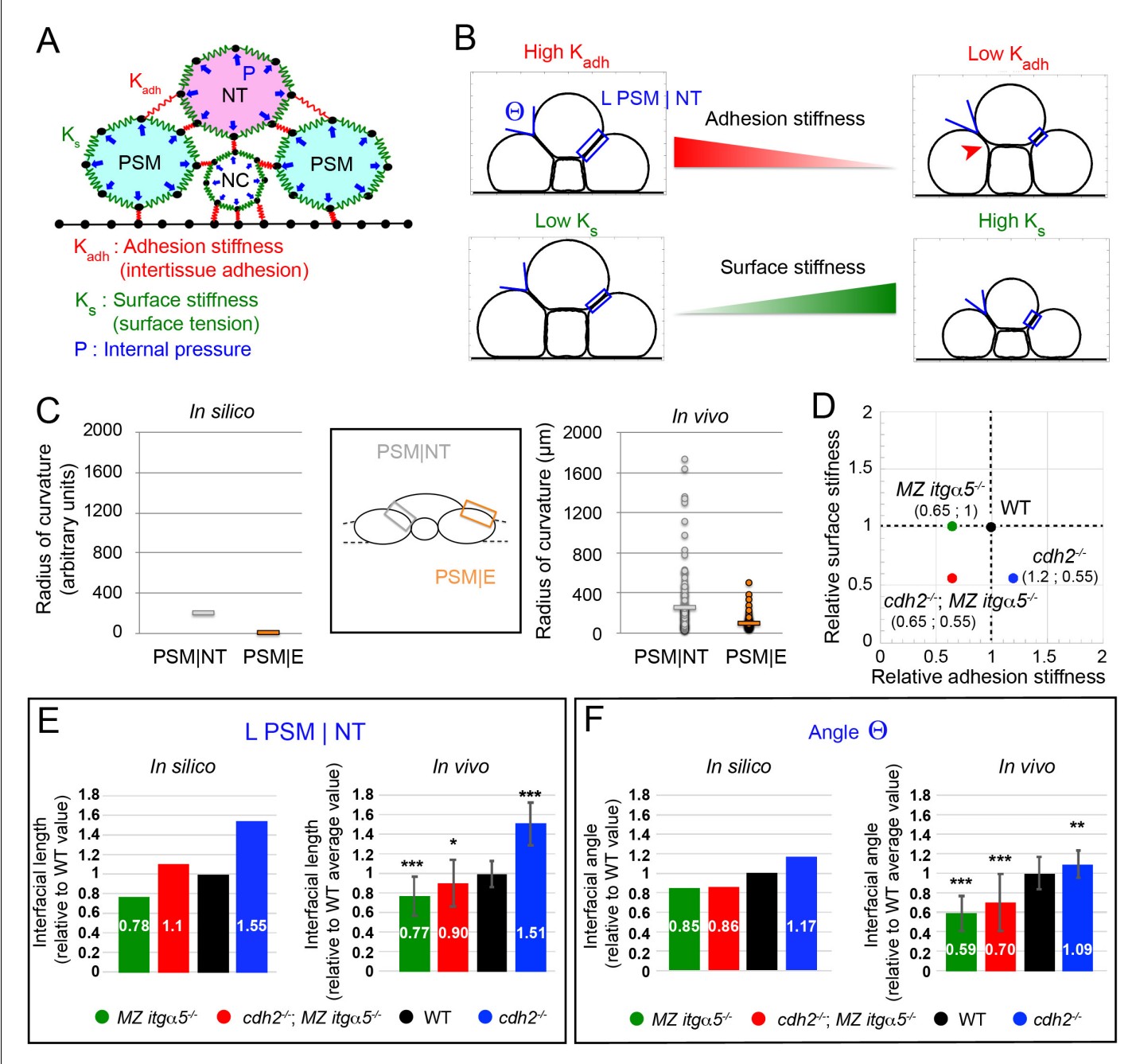

**Figure 2.** A computational model predicts tissue shapes across genotypes. (**A**) A coarse-grained 2D model of a transverse section with three model parameters. Tissues are modeled as soft units with a fixed internal pressure (P, blue arrows), a surface stiffness (green springs along the tissue surfaces with individual spring constant $K_s$), and an adhesion stiffness (red springs connecting adjacent tissues with individual spring constant $K_{adh}$). Black dots represent material points subject to the forces. Black lines at the bottom model a rigid yolk surface. See Materials and methods for further details. (**B**) Variation of adhesion stiffness and surface stiffness in the model have opposing effects on the angle formed at the interface between PSM and neural tube (angle Θ) and on the length of the PSM|NT interface (L PSM|NT, blue box). Decreasing adhesion stiffness decreases interfacial angle and length, while decreasing surface stiffness increases angle and length. Reduction of adhesion stiffness also produces inter-tissue gaps showing local detachments (red arrowhead, top right panel). See Materials and methods for parameter values. (**C**) Radius of curvature for the PSM|NT interface (grey) and the PSM|E interface (orange) in silico (left panel) and in vivo (right panel). n = 150 on 75 sections from five embryos. (**D**) Different genotypes are arranged in a 2D parameter space according to their estimated levels of surface stiffness and adhesion stiffness relative to wild type. See Materials and methods for the details of the parameter values. (**E–F**) Comparison of Interfacial length (L PSM|NT) (**E**) and angle Θ (**F**) across genotypes relative to the average wild-type values, measured in silico and in vivo. Data represent mean with SD. Measurements in vivo were performed within the 140 μm posterior of last somite. n interfacial lengths = 150 on 75 sections on five embryos for WT and *MZ itgα5⁻/⁻*; 123 on 64 sections on five embryos

*Figure 2 continued on next page*

*Figure 2 continued*

for *cdh2^-/-*; 129 on 73 sections on five embryos for *cdh2^-/-*; *MZ itgα5^-/-*. n angles = 145 on 75 sections on five embryos for WT; n = 115 on 63 sections on five embryos for *cdh2^-/-*; 137 on 72 sections on five embryos for *MZ itgα5^-/-*; 133 on 65 sections on five embryos for *cdh2^-/-*; MZ *itgα5^-/-*. T-tests were performed for the following comparisons: *cdh2^-/-* vs WT, p=2.97E-25 (L PSM|NT) p=0.0007 (angle Θ); *MZ itgα5^-/-* vs WT, p=1.16E-13 (L PSM|NT) p=5.37E-30(angle Θ); *MZ itgα5^-/-*;*cdh2^-/-* vs WT, p=0006 (L PSM|NT) p=4.34E-11 (angle Θ).

The online version of this article includes the following figure supplement(s) for figure 2:

**Figure supplement 1.** Parameter exploration of a simple 2D model of tissue morphology.

This simple computational model can account for the gradual narrowing of the neural tube along the medial lateral axis as it develops (*Figure 2—figure supplement 1C*). We can assume that the NT-PSM interface is locally at steady-state along the anterior-posterior axis, and the observed tissue shape change along the anterior-posterior axis may be caused by a progressive increase in K_s along that axis. Indeed, the zebrafish PSM progressively solidifies from posterior to anterior in a *cadherin2*-dependent manner (*Mongera et al., 2018*). We found that the length of the medial to lateral interface between the PSM and NT (L PSM|NT) steadily decreases with increasing surface stiffness when other parameters are fixed (*Figure 2—figure supplement 1A*).

Given the above correspondence between in vivo and in silico observations, we next assigned reasonable values of surface stiffness ($K_S$) and inter-tissue adhesion stiffness ($K_{adh}$) that reproduce morphometrics of wild-type embryos. We note that in vivo the value of the interfacial length (L PSM|NT) decreases to 0.6 of the maximum value at the anterior end of PSM relative to the posterior end (*Figure 2—figure supplement 1C*). We also found in silico that L PSM|NT roughly falls to 0.6 of the maximum value at $K_S \approx 100$ for a fixed $K_{adh}$ (*Figure 2—figure supplement 1A*). Since we are interested in steady-state shapes near the anterior PSM, we may take this value to represent the wild type. Next, we consider the value of $K_{adh}$. In simulations, we varied $K_{adh}$ for different fixed values of $K_S$, and measured the medial-lateral length of NT relative to the interfacial length between NT and PSM (*Figure 2—figure supplement 1G*). In vivo, this ratio (ML length of NT/L PSM|NT) is about two on average for the anterior 140 μm of PSM. In silico, we found that irrespective of the $K_S$ value, this ratio saturates around a value of 2.7 in the range of $K_{adh}$ = 8 to 12 (marked in red in *Figure 2—figure supplement 1G*). Based on the above analysis, we then represented the wild-type embryos by a pair of values: ($K_S,K_{adh}$)=(100,10).

After fixing the wild-type parameters, we then chose the parameter values corresponding to *cdh2* mutants and *MZ itgα5* mutants by matching the in silico and in vivo lengths of PSM|NT interface in these mutants relative to its wild-type values (*Figure 2E*). For *cdh2* mutants, the mean length of PSM|NT is around 1.5 times higher than wild type. To reproduce the experimentally observed increase of L PSM|NT relative to wild-type, we assigned reasonable parameter values to *cdh2* mutants depending on biological expectations. First, we lowered the surface stiffness relative to wild type, since low cell-cell adhesion is known to reduce tissue surface tension (*David et al., 2014*; *Manning et al., 2010*). Second, Cadherin 2 was shown to inhibit Integrin α5 activation and Fibronectin matrix assembly in the PSM (*Jülich et al., 2015*; *McMillen et al., 2016*). Therefore, we may assign a higher adhesion stiffness value to *cdh2* mutants compared to wild type. After systematic exploration of the parameter space in simulations, we found a pair of parameter values, ($K_S,K_{adh}$)=(55,12), that reproduce the in vivo increase of L PSM|NT relative to wild-type (1.55 times).

Following the same procedure for *MZ itgα5* mutants, the in vivo data indicate that *MZ itgα5* mutants exhibit a mean PSM|NT interface length 0.77 times that of the wild-type value. Since cell-matrix interaction is reduced in *MZ itgα5* mutants, it is logical to assume a lower inter-tissue adhesion for this genotype relative to wild type, but the surface stiffness was kept same as wild type. We found that a pair of parameter values, ($K_S,K_{adh}$)=(100,6.5), reproduce the experimentally observed decrease of L PSM|NT relative to wild-type (0.78 times).

Using the parameter values corresponding to *cdh2* mutants and *MZ itgα5* mutants, we can then predict the interfacial length for *cdh2; MZ itgα5* mutants simply by combining these parameter values. The double mutants are represented using the value of inter-tissue adhesion in *MZ itgα5* mutants, and the same value of surface stiffness as *cdh2* mutants (*Figure 2D*). Hence, *cdh2; MZ itgα5* mutants are represented by the values: ($K_S,K_{adh}$)=(55,6.5). These parameter choices for *cdh2* mutants and *MZ itgα5* mutants accurately predicted the in vivo interfacial length of *cdh2; MZ itgα5*

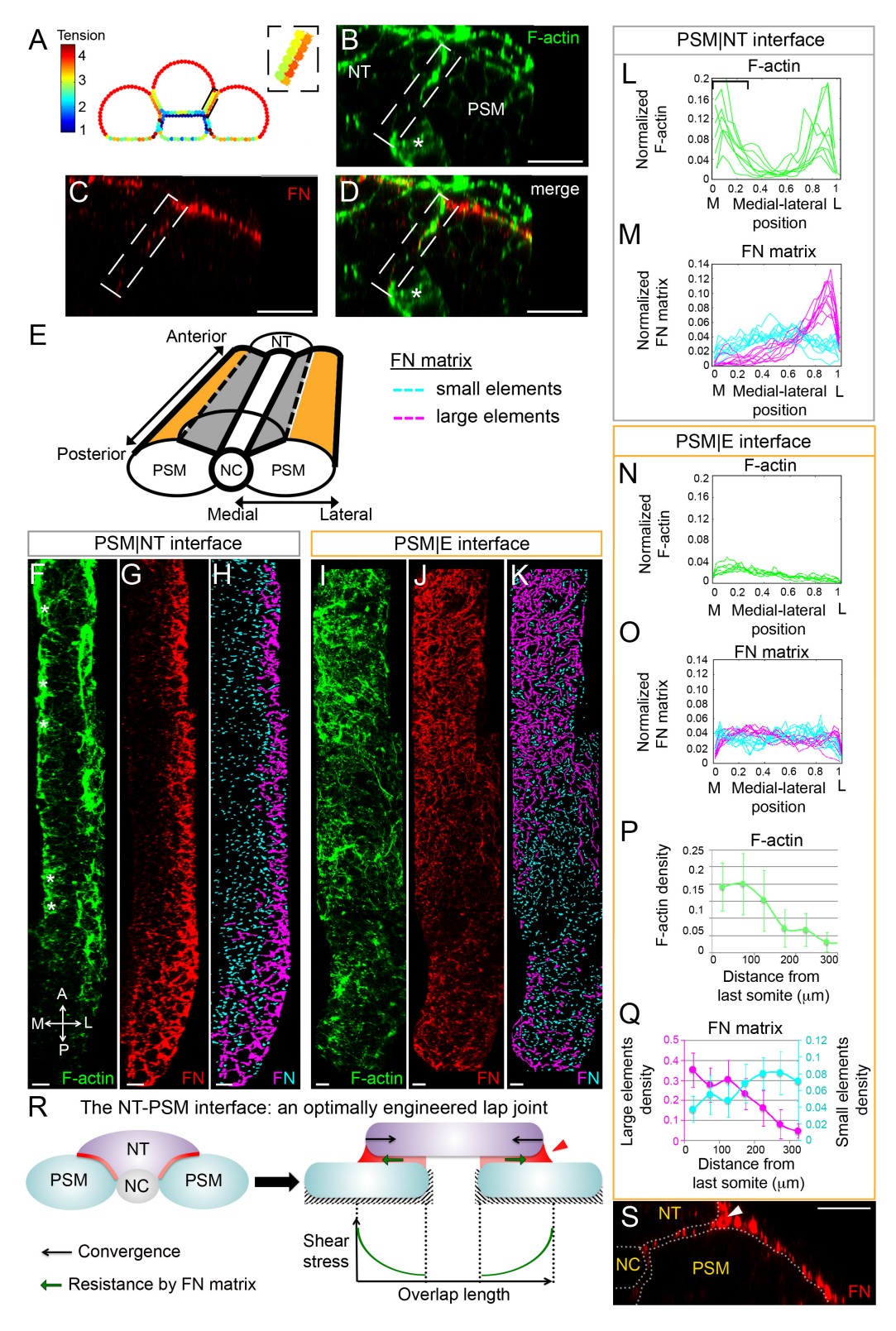

**Figure 3.** Gradients of Fibronectin matrix and F-actin correlate with in silico gradients of tension. (**A**) Heat map of the tension distribution along tissue surfaces in silico. Warmer colors represent higher tension. Inset: magnification of the PSM|NT interface showing a medial-lateral tension gradient on the neural tube side of the interface. (**B–D**) Transverse sections taken 60 μm away from last somite boundary on a 12–14 s stage embryo co-labeled for Fibronectin (FN, red) and F-actin (green). There are colocalized medial-lateral gradients of both F-actin and Fibronectin matrix along the PSM|NT

*Figure 3 continued on next page*

*Figure 3 continued*

interface (dashed box). Star denotes differentiating myofibers that show high F-actin signal. Scale bars = 20 µm. (**E**) 3D schematic of the PSM|NT interface (grey) and the PSM|E interface (orange). To quantify matrix assembly along these interfaces, FN matrix signal is sorted into two categories based on size: small matrix elements (cyan) and large matrix elements (magenta). See SI for details. The F-actin signal (**F, I**), FN signal (**G, J**), and processed images of small and large matrix elements (**H, K**) are shown. The PSM|NT interface exhibits medial-lateral gradients of F-actin and Fibronectin (**F–H**), whereas the PSM|E interface shows an increase in F-actin and Fibronectin matrix assembly along the posterior to anterior axis (**I–K**). All images are projected dorsal views. A = anterior, P=posterior, M = medial, L = lateral. Scale bar = 10 µm. Quantification of the medial-lateral distributions of F-actin (**L**) and small and large FN matrix elements (**M**) along the PSM|NT interface. The bracket in (**L**) denotes the differentiating myofibers rich in F-actin. Quantification of the medial-lateral distributions of F-actin (**N**) and small and large FN matrix elements (**O**) along the PSM|E interface. Quantification the density of F-actin (**P**) and small and large matrix elements (**Q**) along the anterior-posterior axis of the PSM|E interface. Data represent means and SD. Sample sizes: L, n = 8 PSMs from six embryos; N, P, n = 7 PSMs from five embryos; M, O, Q, n = 10 PSMs from six embryos. (**R**) The NT-PSM interfaces represented as a lap-joint with a single sided strap. Neural tube (magenta) acts as a strap that is adhered to the two PSMs (cyan) via a graded adhesive (red) made of Fibronectin. Medial and ventral edges of PSM are attached to the notochord and yolk surface respectively (dashed region). Black arrows denote neural tube convergence and green arrows denote resistance to this convergence via the adhesive. Neural tube convergence with respect to the adhesive produces shear stress. Established theories of lap-joint predict a stress gradient with higher stress at the lateral edge of the PSM|NT interface. Extra adhesive, called a 'spew fillet,' in an arced shape at the lateral sides of the strap strengths the joint. (**S**) Transverse section on a 12–14 s stage embryo labeled for Fibronectin (FN, red) illustrating the spew fillet of FN matrix (arrowhead). Scale bar = 20 µm. The online version of this article includes the following figure supplement(s) for figure 3:

**Figure supplement 1.** Medial-lateral gradients of Myosin II and Fibronectin matrix tension at the PSM|NT interface.

**Figure supplement 2.** Epithelialization of PSM surface cells and increases in F-actin intensity in PSM cells as the PSM matures from posterior to anterior.

---

mutants, (*Figure 2E*). Importantly, although we assigned the parameter values depending on the relative interfacial lengths of *cdh2* mutants and *MZ itgα5* mutants, these parameter choices also reproduced the experimentally observed trends in the variation of interfacial angle Θ across genotypes (*Figure 2F*).

## The neural tube and PSM are linked via an adhesive lap-joint with gradients of F-actin, Myosin-II and Fibronectin matrix

The observation that the PSM|NT and PSM|E interfaces have very different curvatures intuitively suggest that they are under different levels of tension. Therefore, we analyzed the tension distributions at tissue surfaces in our simulations (*Figure 3A*). Interestingly, the model predicted a gradient of tension at the neural tube side of PSM|NT interface with a higher tension on the lateral side of the interface and a lower tension on the medial side (see dashed box, *Figure 3A*). In contrast, the medial-lateral tension distribution at the PSM|E surface is predicted to be homogeneous.

To test these predictions in vivo, we analyzed F-actin intensity as an indicator of cortical tension. Consistent with the model prediction, we observed an increasing medial to lateral gradient of F-actin along the PSM|NT interface (*Figure 3B, F and L*). A parallel gradient of Myosin-II localization was also observed (*Figure 3—figure supplement 1A–1C*). Fibronectin matrix assembly is a force-induced process dependent upon cell actomyosin contractility (*Zhong et al., 1998*; *Lemmon et al., 2009*; *Dzamba et al., 2009*). We therefore analyzed the Fibronectin matrix to determine whether there was a correlation between levels of F-actin and Fibronectin matrix assembly at the tissue interfaces (*Figure 3C and D*). Since Fibronectin matrix is assembled from soluble and small matrix aggregates into large fibers, we quantified the topology of the Fibronectin matrix by sorting into small and large matrix elements, color coded in cyan and magenta, respectively (*Figure 3H, K, M and O*). In correlation with the F-actin gradient, we observed a medial to lateral gradient of Fibronectin matrix assembly as large assembled elements were enriched at lateral edge of the interface, while small matrix elements were evenly dispersed. Lastly, we used a monoclonal antibody that recognizes an epitope in human Fibronectin that is exposed when Fibronectin is under tension (*Cao et al., 2017*). In embryos expressing a chimeric zebrafish/human Fibronectin, we observed a medial to lateral gradient of Fibronectin under tension along the PSM|NT interface (*Figure 3—figure supplement 1D–1F, J and K*). All together, these data support the model prediction that there is an increasing medial to lateral gradient of tension along the interface between the PSM and neural tube.

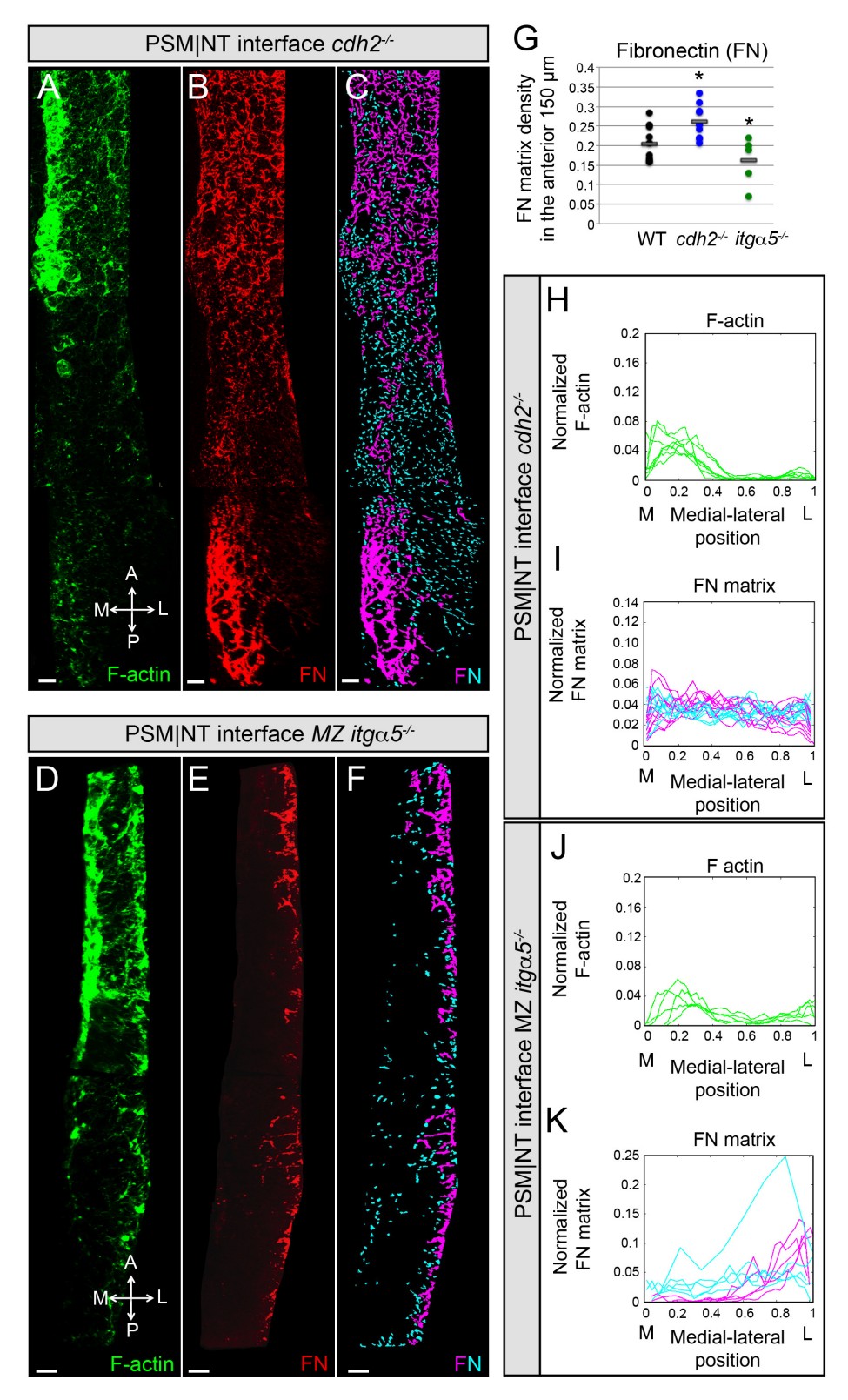

**Figure 4.** Reduction of cell-cell adhesion eliminates the medial-lateral gradients of Fibronectin matrix and F-actin. PSM|NT interfaces of *cdh2-/-* (A, B, C) and *MZ itgα5-/-* (D, E, F) mutants at the 12–14 somite stage. F-actin (A, D), Fibronectin (B, E), and processed images of small and large Fibronectin matrix elements (C, F) are shown. All images are dorsal views. A = anterior, P = posterior, M = medial, L = lateral. Scale bars = 10 μm. (G) Quantification of Fibronectin density within the anterior 150 μm of the PSM|NT interface in wild type, *cdh2-/-* and *MZ itgα5-/-*. *cdh2-/-* vs WT, p=9.4e-3;

*Figure 4 continued on next page*

Figure 4 continued

*MZ itgα5⁻/⁻* vs WT, p=0.012. Quantification of the medial-lateral distributions of the F-actin (**H, J**) and small and large Fibronectin matrix elements (**I, K**) at the PSM|NT interface in *cdh2⁻/⁻* (**H, I**) and *MZ itgα5⁻/⁻* (**J, K**). Sample sizes: G, I, K, n = 10 PSMs from six embryos for WT, n = 10 PSM from five embryos for *cdh2⁻/⁻*, n = 5 PSM from five embryos for *MZ itgα5⁻/⁻*. Sample sizes: H, n = 7 PSM from four embryos; J, n = 5 PSM from five embryos.

These medial-lateral gradients of F-actin and Fibronectin matrix were only observed at the PSM|NT interface. Both F-actin, Fibronectin matrix assembly and Fibronectin under tension were homogenously distributed medial-laterally at the PSM|E interface (*Figure 3B–O* and *Figure 3—figure supplement 1G–1I and L–M*), which is again consistent with the model predictions. Instead, this interface was characterized by a progressive posterior to anterior increase in F-actin density (*Figure 3I and P*). Similarly, we observed a posterior to anterior increase in the density of large matrix elements and decrease in the density of small matrix elements (*Figure 3K and Q*). These opposing gradients suggest that the matrix is progressively assembled from posterior to anterior by crosslinking small Fibronectin fibrils to form large fiber networks (*Figure 3J*). This posterior-to-anterior increase in matrix density and F-actin at the PSM|E interface also correlates with the progressive epithelialization of PSM surface cells and a posterior to anterior increase of F-actin signal in both internal cells and surface cells of the PSM (*Figure 3—figure supplement 2*).

Overall, the intriguing observation here is that the homogeneous medial-lateral distribution in F-actin and Fibronectin density observed at the PSM|E interface is lost at the PSM|NT interface, which instead displays a medial-lateral gradation both in F-actin and the Fibronectin fibers. Our simplified model suggests that this behavior may result from the particular distribution of mechanical stress at the PSM|NT interface. This interface between two converging tissues mechanically resembles an 'adhesive lap joint' that is commonly used in engineering and is comprised of partially overlapping components bound via an adhesive. The structure formed by the apposition of neural tube and PSM tissues can be idealized as a particular type of lap joint, a single-sided strapped joint (*Ghoddous, 2017*), where the neural tube on top of the PSM acts as a single sided strap bridging the two PSMs via a Fibronectin adhesive (*Figure 3R*). Shear stress at the overlapping interface of a lap joint is highest at the lateral edges of the overlap and lowest in the middle (*Goland and Reissner, 1944*). This nonhomogeneous stress distribution can be compensated for by grading the adhesive in the overlap (*Carbas et al., 2014*). In accordance to this engineering insight, we observe a graded Fibronectin 'adhesive' at the PSM|NT interface. An important distinction is that shear stress is generated by an externally applied load in an engineered lap joint, while at the PSM|NT tissue interface, shear stress is generated via tissue motion relative to one another due to convergence extension. This shear stress at the adhesive interface resists the convergence of neural tube (green and black arrows in *Figure 3R*, right). In engineering, excess adhesive can be added to the edges of the overlap and sculpted into an arc to strengthen the joint by distributing the forces over a larger area (*Figure 3R*; *Lang and Mallick, 1998*). These arced adhesive 'spew fillets' are also observed in our system at the lateral edges of the PSM|NT interface (*Figure 3S*). In addition, the geometry of the adherent materials impacts the strength of a lap joint. A rounded end (rather than an end with sharp corners) of adherent materials eliminates singularities in the stress field and strengthens the joint by more evenly distributing the forces (*Adams and Harris, 1987*). Here, the tissue edges are rounded at PSM|NT interface. Thus, the PSM|NT interface behaves like an optimally engineered adhesive lap joint with rounded edges, a graded adhesive and an adhesive spew fillet.

## Effects of decreases in cell-cell adhesion and cell-matrix interaction on the PSM|NT interface

In our computational model, we hypothesized a greater inter-tissue adhesion between the neural tube and the PSM in *cdh2* mutants and a lower inter-tissue adhesion in *itgα5* mutants compared to wild-type embryos. These assumptions were sufficient to accurately predict the relative in vivo values of both the interfacial angle and length of PSM|NT interface. To further test these model assumptions, we quantified Fibronectin matrix assembly and F-actin at PSM|NT interface in *cdh2* mutants and *MZ itgα5* mutants. We observed relatively high average levels of Fibronectin at the interface in the *cdh2* mutants and the gradients of both Fibronectin matrix assembly and F-actin were lost (*Figure 4A–C and G–I*). In contrast, *MZ itgα5* mutants have lower levels of Fibronectin, and the medial-lateral gradients of F-actin and Fibronectin matrix assembly were maintained (*Figure 4D–K*).

Lap joints exhibit graded shear stress at the interface of two overlapping materials (*Goland and Reissner, 1944*). In our system, the presence of a shear stress at the interface should depend on the relative motion between the attached converging tissues, and this relative motion can be reduced in the absence of neural tube convergence. If the medial-lateral gradation of Fibronectin matrix assembly and F-actin are shear-driven, we can expect a positive correlation between neural tube convergence and presence of these gradients. Our data for *cdh2* mutants strengthen this idea as these mutants exhibit a reduction in neural tube convergence as well as loss of the gradients. On the other hand, *itgα5* mutants do not have a defect in convergence and they retain the potential to establish this graded assembly, though we observe an overall reduction in Fibronectin matrix production.

## Fibronectin is required for inter-tissue adhesion

To directly examine whether Fibronectin matrix is required for inter-tissue adhesion and restriction of neural tube convergence, we generated double mutants for the two zebrafish *fibronectin* genes, *fn1a* and *fn1b*, using CRISPR/Cas9 (*Figure 5—figure supplement 1B*). Double homozygous fibronectin mutant embryos (*fn1a⁻/⁻;fn1b⁻/⁻*) completely lacked Fibronectin matrix as detected by immunohistochemistry (IHC) (*Figure 5D*). *fn1a⁻/⁻;fn1b⁻/⁻* embryos exhibited a precociously converged neural tube, local tissue detachments, and displayed local disruptions of bilateral symmetry consistent with the *MZ itgα5⁻/⁻* phenotype (*Figure 5A and B*, *Figure 5—figure supplement 1C–1E*).

We created a photoconvertible Fibronectin transgenic to enable us to 'paint' the extracellular matrix and quantify deformation of the 'painted spots' in live embryos. The transgenic line expresses Fibronectin 1a tagged with the green-to-red photoconvertible protein mKikumeGR under the control of the heat-shock promoter (*Tg hsp70:fn1a-mKIKGR*, *Figure 5—figure supplement 1A*). We tested the functionality of the transgene by creating *fn1a⁻/⁻;fn1b⁻/⁻; hsp70:fn1a-mKIK* embryos. *fn1a⁻/⁻;fn1b⁻/⁻* embryos are not viable, and we observed somite border defects consistent with

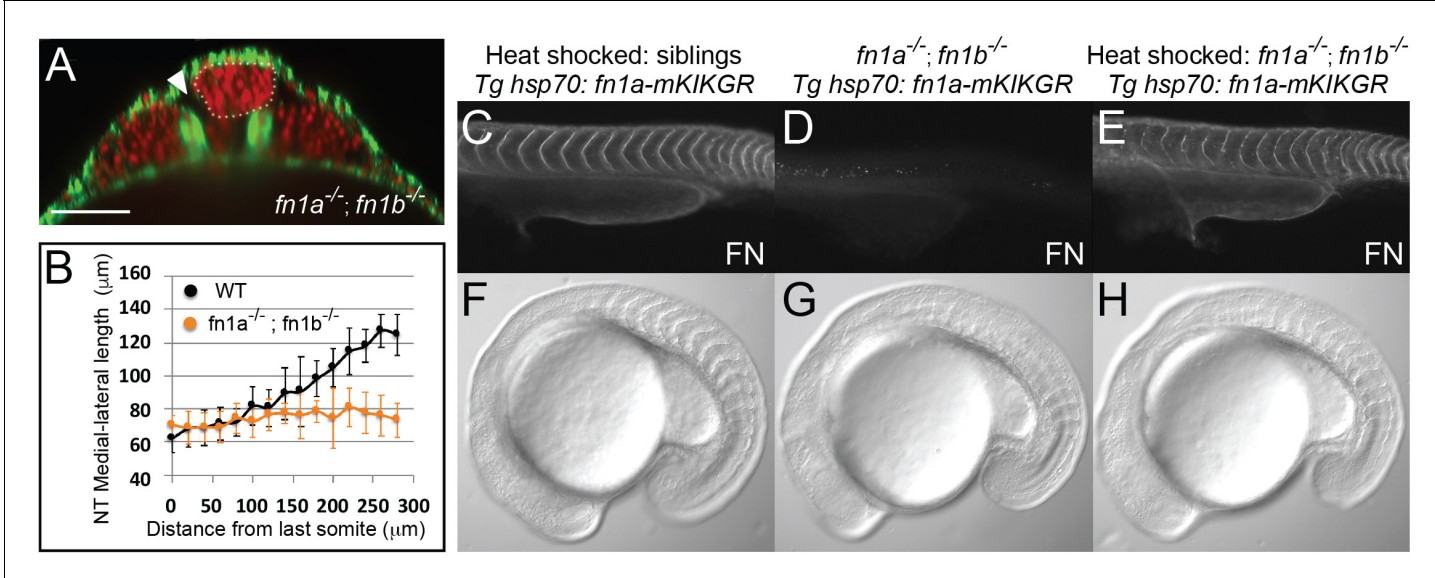

**Figure 5.** Fibronectin is required for inter-tissue adhesion and a *hsp70: fn1a-mKIKGR* transgene rescues *fn1a⁻/⁻; fn1b⁻/⁻* double mutants. (**A–B**) *fn1a⁻/⁻; fn1b⁻/⁻* double mutants exhibit a precociously converged neural tube and tissue detachment similar to *MZ itgα5⁻/⁻* mutants. (**A**) Transverse section 160 μm from last somite boundary in 12–14 somite stage *fn1a⁻/⁻; fn1b⁻/⁻* embryos with labeled nuclei (red) and F-actin (green). Dotted line delineates the neural tube. Arrowhead indicates tissue detachment. Scale bar = 70 μm. (**B**) Quantification of the medial-lateral length of the neural tube along the anterior-posterior axis starting from the last somite boundary (0 μm). Data represent means and SD. Immunostaining for Fibronectin (FN) on a 24 hpf heat shocked sibling (**C**), a non-heat shocked *fn1a⁻/⁻; fn1b⁻/⁻; Tg hsp70: fn1a-mKIKGR* embryo (**D**) and a heat shocked *fn1a⁻/⁻; fn1b⁻/⁻; Tg hsp70: fn1a-mKIKGR* embryo (**E**). Lateral views of the trunk with anterior to the left. DIC images of a heat shocked sibling (**F**), a non-heat shocked *fn1a⁻/⁻; fn1b⁻/⁻; Tg hsp70: fn1a-mKIKGR* embryo (**G**) and a heat shocked *fn1a⁻/⁻; fn1b⁻/⁻; Tg hsp70: fn1a-mKIKGR* embryo (**H**).

The online version of this article includes the following figure supplement(s) for figure 5:

**Figure supplement 1.** Generation of *Tg hsp70: fn1a-mKIKGR* transgenic zebrafish to study matrix remodeling in live embryos and generation of double *fibronectin* mutants which exhibit precocious neural tube convergence and left-right PSM asymmetries.

published loss of function studies in zebrafish (*Figure 5G*; *Jülich et al., 2005*; *Koshida et al., 2005*). Heat-shocked *fn1a⁻ᐟ⁻;fn1b⁻ᐟ⁻; hsp70:fn1a-mKIK* embryos exhibited normal Fibronectin assembly and somite boundary morphologies, demonstrating the functionality of the transgene (*Figure 5C–H* and *Tables 1* and *2*). Using this transgene, we compared the dynamics of matrix remodeling at different tissue interfaces in wild-type and mutant embryos.

## Medial-lateral matrix remodeling at the PSM|NT interface depends on neural tube convergence and tissue attachment

We assayed Fibronectin matrix dynamics at tissue interfaces in vivo via confocal timelapse microscopy. When transgenic embryos were heat-shocked at the 2–3 somite stage, we observed a fluorescently labeled Fibronectin matrix at the 12-somite stage (*Figure 6B and C*). The topology of the Fn1a-mKikGR matrix was consistent with the matrix topology observed at each interface in fixed wild-type samples subjected to Fibronectin IHC. We photoconverted spots of matrix 25–35 μm in diameter either at the PSM|NT or PSM|E interface (*Figure 6A–C*) at a distance of 150–200 μm from the last somite, and imaged every 15 min. One hour after the photoconversion, the spots at the PSM|NT interface showed significant shrinking along the medial-lateral direction, while the spots at the PSM|E interface did not shrink (*Figure 6B and C* and *Video 2*).

We quantified the spatiotemporal dynamics of photoconverted Fibronectin matrix spots after thresholding to account for photobleaching. The spots were converted into binary images and, we used the standard deviation of the white pixel distribution along the medial-lateral and anterior-posterior axes as measures of the width and height, respectively. This metric quantifies the general deformation of the photoconverted spot over time. At the PSM|NT interface, the medial-lateral width of the photoconverted region decreased by 40%, while the anterior-posterior height was unchanged (*Figure 6D* and *Figure 6—figure supplement 1A*). In contrast, the matrix at the PSM|E interface did not exhibit medial-lateral shrinking (*Figure 6D* and *Figure 6—figure supplement 1A*). Further analysis of the directionality of matrix remodeling revealed that there is medial to lateral bias in the displacement fields of the PSM|NT interface but no anisotropy in the displacement fields of the PSM|E interface (*Figure 6—figure supplement 1E–1G*).

Lap joint theory suggests that inter-tissue adhesion is required for shear forces at the PSM|NT interface. We examined the effect of inter-tissue adhesion on shear driven Fibronectin matrix remodeling utilizing the variable inter-tissue detachment phenotype of *MZ itgα5* mutants in three groups of photoconversion experiments (*Figure 6E and F*). In Group 1 embryos, the neural tube is attached on both sides to the PSM at the level of the photoconverted region (*Figure 6G*). In Group 2 embryos, tissues are detached ipsilateral to the photoconverted side but attached on the contralateral side (*Figure 6H*). In Group 3 embryos, tissues are attached ipsilateral to the photoconverted side but detached on the contralateral side (*Figure 6I*). In the absence of tissue detachment, the photoconverted matrix was remodeled in the medial-lateral direction similar to wild type (*Figure 6G and J*). In contrast, where tissues were ipsilaterally detached the medial-lateral remodeling was lost (*Figure 6H and J*). This result demonstrates that inter-tissue adhesion is necessary for the medial-lateral matrix remodeling. Strikingly, tissue detachment contralateral to the photoconverted region also affected matrix remodeling (*Figure 6I and J*). This contralateral phenotype illustrates a long-range effect of mechanical coupling, namely that ECM remodeling on one side of the embryo is responsive to morphogenic variability on the opposite side of the embryo.

We next asked whether the medial-lateral remodeling of the Fibronectin matrix at the PSM|NT interface was dependent on neural tube convergence by performing the photoconversion experiment in *cdh2* mutants. These mutants exhibit variable neural tube convergence as measured by the change in medial-lateral width of the neural tube over time. We also observed bilaterally asymmetric convergence of the neural tube in some embryos, which we quantified as the change in medial-lateral width of the left and right halves (*Figure 6M*). We divided *cdh2* mutants into three groups based the variable neural tube convergence phenotype. In Group 1, the neural tube is symmetric and slowly converges at the level of the photoconverted region (*Figure 6K and N*). In Group 2, the neural tube is symmetric at the level of the photoconverted region but convergence is severely retarded (*Figure 6L and N*). In Group 3, the neural tube converges asymmetrically at the level of the photoconverted region (*Figure 6M and P*). When the neural tube converges symmetrically, the matrix exhibits normal medial-lateral remodeling (*Figure 6K and O*). However, when the neural tube

**Table 1.** Rescue experiments with a heat shock *hsp70: fn1a-mKikGR* at shield stage.

For each experiment, *fn1a-/+; fn1b-/+;hsp70: fn1a-mKikGR* adults were crossed. Three crosses include a parent homozygous for the transgene while one cross was from parents both hemizygous for the transgene. Embryos from each clutch were divided in half. 50% of embryos were controls that were not heat shocked, and 50% of embryos were heat shocked at the shield stage. The numbers shown correspond to the total number of embryos presenting each phenotype (either wild-type, *fn1a-/-* or *fn1a-/-; fn1b-/-*) based on the presence of somite border defects at the 14–18 somite stage.

| | Embryos with no heat shock | Heat shocked embryos | |
| --- | --- | --- | --- |
| | | Fluorescent | Non-fluorescent |
| Phenotypically wild-type | 301 | 351 | 26 |
| *fn1a-/-* | 91 | 0 | 3 |
| *fn1a-/-; fn1b-/-* | 20 | 0 | 4 |

does not converge, there is no medial-lateral matrix remodeling (*Figure 6L and O*). Thus, anisotropic matrix remodeling at the PSM|NT interface depends on the neural tube convergence.

An interesting phenotype was observed when the neural tube converges asymmetrically. The side that exhibited a strong neural tube convergence defect displayed normal medial-lateral matrix remodeling, while anisotropic matrix remodeling was lost on the contralateral side where the neural tube converged (*Figure 6M and Q*). This result indicates that neural tube convergence on one side of the body impacts the Fibronectin matrix remodeling on the opposite side of the body.

We next performed the photoconversion experiments in *cdh2; MZ itgα5* double mutants. Mutant embryos were sorted into two groups based on the degree of neural tube convergence. We observed medial-lateral Fibronectin matrix remodeling when the neural tube converged, whereas medial-lateral matrix remodeling was lost with diminished neural tube convergence (*Figure 6R–U*). Thus, these experiments confirmed that the medial-lateral remodeling of the matrix positively correlates with the neural tube convergence. Lastly, we did not observe any change in the anterior-posterior length of the photoconverted spots for any mutants, similar to wild type (*Figure 6—figure supplement 1B–1D*).

We revisited these contralateral effects with our in silico model using different parameter values for the left and right sides. Uniform parameters produce symmetric gradients in tension in the neural tube at the left and right PSM|NT interfaces (*Figure 7A*). When the adhesion stiffness (representing inter-tissue adhesion) is reduced along the left interface relative to the right interface, the left and right PSM become asymmetric in size and the contralateral tension gradient becomes shallower (*Figure 7B*). This result mimics both the morphological phenotype as well as the contralateral effects on Fibronectin matrix remodeling observed in *MZ itgα5-/-* embryos. We computationally implemented asymmetric neural tube morphogenesis observed in some *cdh2* mutants by reducing the

**Table 2.** Rescue experiments with heat shock *fn1a-mKikGR* at the 10–12 somite stage on pre-sorted *fn1a-/-; fn1b-/-* embryos.

For each experiment, *fn1a-/+; fn1b-/+;hsp70: fn1a-mKikGR* adults were crossed and embryos sorted for the *fn1a-/-; fn1b-/-* morphological phenotype were heat shocked at the 10–12 stage and assayed for somite border defects at 24 hpf. 1 or two embryos per experiment were not heat shocked as a control. Four experiments were performed and the numbers shown in the table represent the total number of embryos with each phenotype. * HS at the 12-somite stage will rescue body elongation and head development defects observed in the *fn1a-/-; fn1b-/-* embryos, however heat shock will not fully rescue border defects of the first 1–6 somites.

| | Embryos with no heat shock | Heat shocked embryos | |
| --- | --- | --- | --- |
| | | Fluorescent | Non-fluorescent |
| Rescued phenotype * | - | 13 | 0 |
| *fn1a-/-; fn1b-/-* | 5 | 0 | 1 |

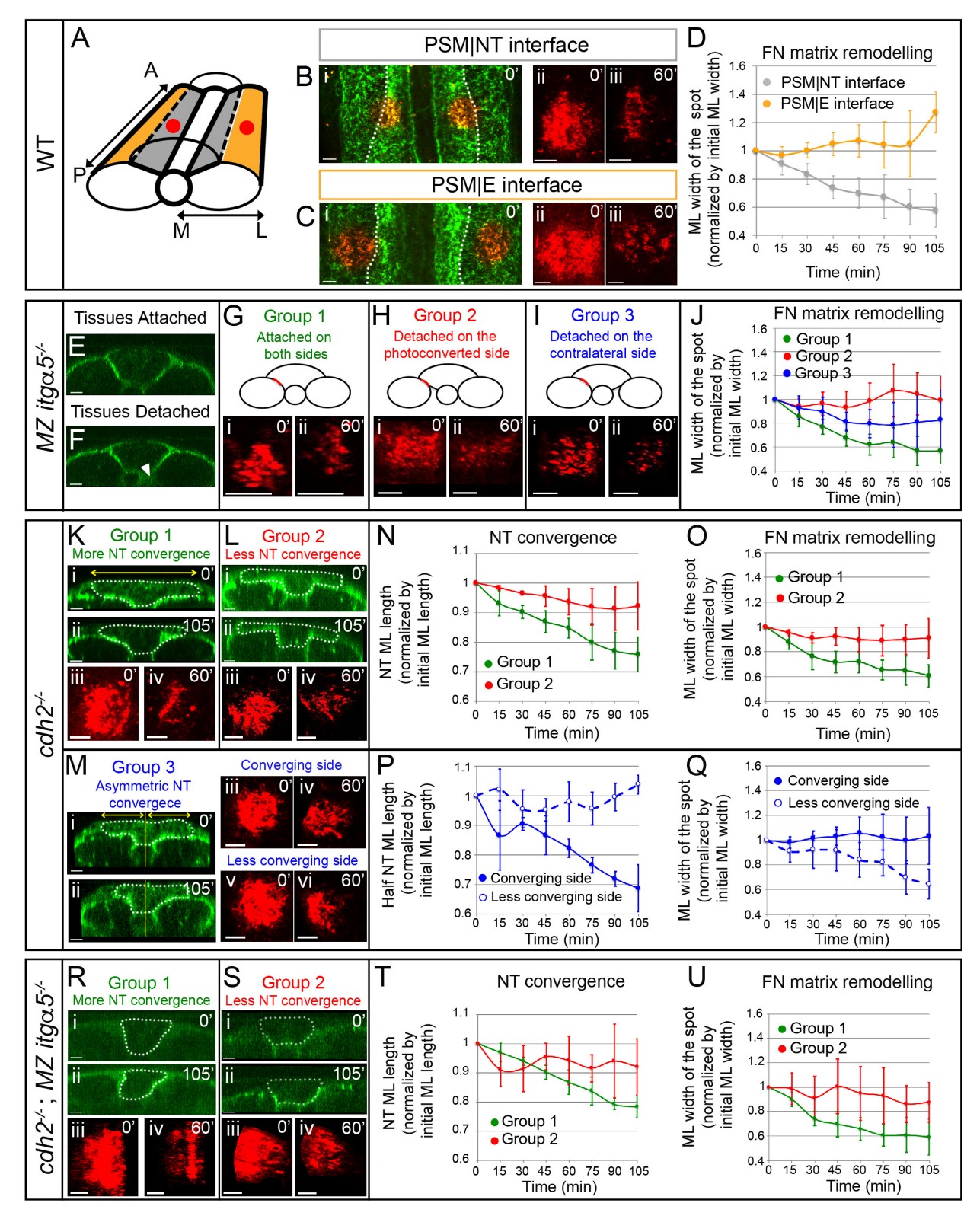

**Figure 6.** Anisotropic Fibronectin matrix remodeling dependent upon neural tube convergence and inter-tissue adhesion. (**A**) 3D schematic indicating positions of the photoconverted regions (red dots) along the PSM|NT and PSM|E interfaces. A = anterior; P = posterior; M = medial; L = lateral. Spots of photoconverted Fibronectin matrix at PSM|NT interface (**B**) or at PSM|E interface (**C**). (i) Dorsal views of heat-shocked *Tg hsp70:fn1a-mKIKGR* embryos at 12 somite stage showing Fibronectin matrix (green) and Fibronectin spots immediately after photoconversion (red). Dotted lines delineate
*Figure 6 continued on next page*

*Figure 6 continued*

the boundary between the PSM|NT and PSM|E interfaces. Images of spots of Fibronectin at early (ii) and late (iii) timepoints. (**D**) Quantification of the medial-lateral width of the photoconverted spot over time. Sample size: n = 9 spots from four embryos for each interface. (**E–J**) Analysis of FN matrix remodeling at PSM|NT interface in *MZ itgα5$^{-/-}$*. (**E–F**) Transverse sections of heat-shocked 12 somite stage MZ *itgα5$^{-/-}$*; *Tg hsp70:fn1a-mKIKGR* embryos. *MZ Itgα5* mutants have local tissue detachments (arrowhead in F). (**G–I**) Photoconversion experiments fall into three groups based on tissue detachment at the level of the photoconverted interface (red line in schematics). (i–ii) Images of the photoconverted spots at the beginning of the movie (i) and one hour later (ii). (**J**) Quantification of the medial-lateral width of the photoconverted spot over time. Sample size: n = 5 spots from three embryos for each of group 1 and 2; 7 spots from four embryos for group 3. (**K–Q**) Analysis of FN matrix remodeling at PSM|NT interface in *cdh2$^{-/-}$*. (**K–M**) *Cdh2* mutants show variability in neural tube convergence and fall into three groups based on the degree of neural tube convergence at the level of the photoconverted interface. Either the neural tube converged (Group 1 (**K**)); did not converge (Group 2 (**L**)); or converged asymmetrically (Group 3 (**M**)). (i–ii) Transverse sections of *cdh2$^{-/-}$*; *Tg hsp70:fn1a-mKIKGR* heat-shocked embryos at the beginning (i, 12 somite stage) and at the end of the time lapse (ii). Dotted lines delineate the neural tube contour. Yellow lines indicate the midline of the embryo used as reference to divide the neural tube in left and right halves. Yellow double arrows indicate how the medial-lateral width of the total neural tube or half of the neural tube was quantified. (iii-vi) Images of the photoconverted spots at the beginning of the movie (iii, v) and one hour later (iv, vi). (**N, P**) Quantification of the medial-lateral length of the total neural tube (**N**) or of the half neural tube (**P**) over time. Sample size: group 1, four embryos; group 2, four embryos; group, three embryos. One transverse section per embryo. (**O, Q**) Quantification of the medial-lateral width of the photoconverted region over time. Sample size: group 1, n = 8 spots from four embryos; group 2, 10 spots from three embryos; group 3, 3 spots from three embryos. (**R–U**) Analysis of FN matrix remodeling at the PSM|NT interface in *cdh2$^{-/-}$*; *MZ itgα5$^{-/-}$* embryos. (**R, S**) Embryos were sorted into two categories based on a high (Group 1) or low (Group 2) degree of neural tube convergence at the level of the photoconverted interface. (i–ii) Transverse sections of *cdh2$^{-/-}$*; *MZ itgα5$^{-/-}$*; *hsp70:fn1a-mKIKGR* heat-shocked embryos at the beginning (i, 12 somite stage) and at the end of the time lapse (ii). Dotted lines delineate the neural tube contour. (iii-iv) Images of the photoconverted spots at the beginning of the movie (iii) and 1 hr later (iv). (**T**) Quantification of the medial-lateral length of the neural tube over time. Sample sizes: n = 3 neural tube sections on two different embryos for each of groups 1 and 2. (**U**) Quantification of medial-lateral width of the photoconverted region over time. Sample sizes: n = 8 spots from three embryos for group 1; 4 spots from two embryos for group 2. All images of photoconverted spots are projected dorsal views perpendicular to the interfaces with anterior to the top and medial to the left. Scale bars of all images = 15 µm. In all plots, dots represent means and error bars represent SD.

The online version of this article includes the following figure supplement(s) for figure 6:

**Figure supplement 1.** Medial-lateral ECM remodeling along the PSM|NT interface.

surface stiffness of one half of the neural tube (*Figure 7C*). Here, we observed a contralateral reduction in the tension gradient which mimics the contralateral effects on Fibronectin matrix remodeling observed in *cdh2$^{-/-}$* embryos.

## Discussion

Here, we find that inter-tissue adhesion between the neural tube and left and right paraxial mesoderm ensures bilaterally symmetric morphogenesis. However, this inter-tissue adhesion predisposes the neural tube to convergence defects and spina bifida by requiring the neural tube to pull on the adjacent mesoderm (*Figure 7D*). We find that the mechanics of this inter-tissue adhesion resemble an adhesive lap joint which is well described in engineering. Fibronectin functions as the adhesive in this lap joint. Due the cellular mechanisms that regulate Fibronectin matrix assembly and remodeling, we find that Fibronectin responds to a gradient of stress by remodeling to the region of highest stress. Thus, Fibronectin functions as a 'smart adhesive' that continually remodels to where it is most needed.

Multiple lines of evidence suggest that the interface between the neural tube and PSM

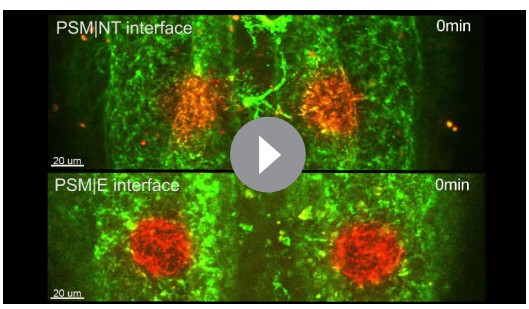

**Video 2.** Tracking Fibronectin matrix dynamics at tissue interfaces. Movies representing 105 min time-lapses (15 min interval) after local photoconversion (red spots) of Fn1a-mKikGR matrix (green) at 12 somite stage, either at the PSM|NT interface (top movie) or at the PSM|E interface (bottom movie). Dorsal views with anterior to the top.
https://elifesciences.org/articles/48964#video2

behaves as an optimally engineered adhesive lap joint. Mutants that reduce or eliminate the Fibronectin matrix exhibit a precociously converged neural tube implicating cell-Fibronectin matrix adhesion in constraining neural tube convergence. In vivo testing of predictions of a computational model suggests that the PSM-neural tube interface is under a medial-laterally graded stress. This stress gradient is characteristic of a lap joint (*Goland and Reissner, 1944*). The medial to lateral remodeling of Fibronectin produces a graded adhesive which strengthens the lap joint by accounting for the graded stress (*Carbas et al., 2014*). In addition, there is an arc-shaped domain of Fibronectin matrix at the rounded lateral edges of the PSM-neural tube interface. This domain of Fibronectin matrix is reminiscent of an 'adhesive spew fillet', which is an excess of adhesive present at the edges of a lap joint that fortifies the joint (*Lang and Mallick, 1998*). Moreover, the rounded edges of the tissues strengthen the lap joint by more evenly distributing stress (*Adams and Harris, 1987*).

The extracellular matrix (ECM) can act both as a mediator of mechanical forces and a stress sensor. The latter function arises by the activation of biochemical signaling pathways via Integrins in response to changes in ECM fiber morphology (*Vogel and Sheetz, 2009*). Moreover, Fibronectin fibrillogenesis is in a positive feedback loop with actomyosin contractility leading to colocalization of

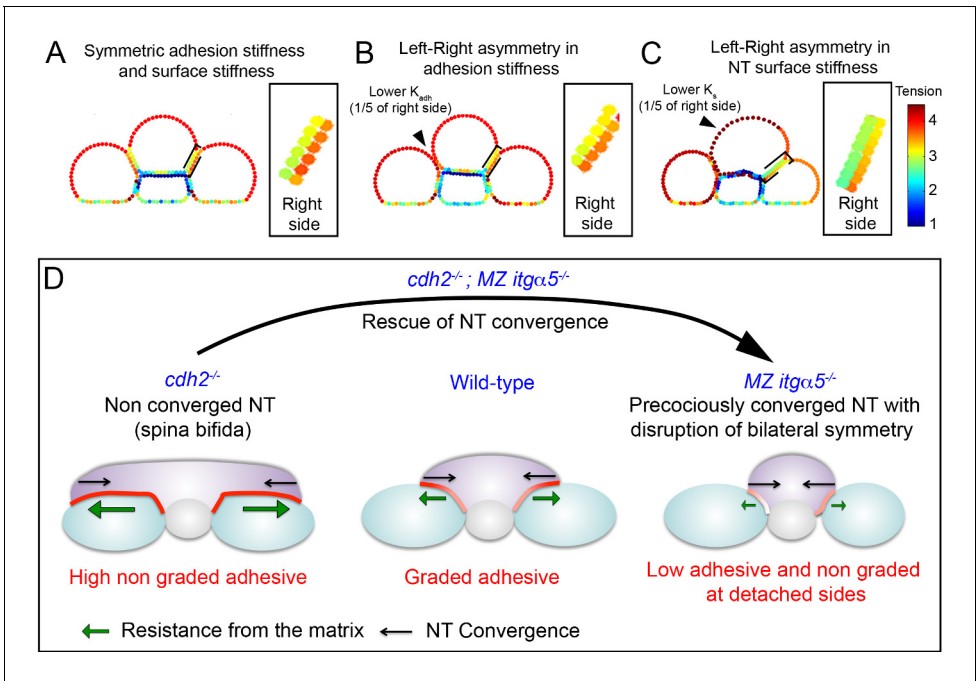

**Figure 7.** Fibronectin mediated inter-tissue adhesion ensures bilaterally symmetric morphogenesis but predisposes the neural tube to convergence defects. (A–C) In silico heat maps of the tension distributions along the tissue surfaces with left-right variation either in adhesion stiffness (B) or in surface stiffness (C), compared to a symmetric condition (A). Warmer colors represent higher tension. Insets: magnification of the right PSM|NT interface, which is unperturbed in every condition, to show the contralateral effects in tension gradients. The gradients at the neural tube side of the interface become shallower in B and C compared to A. (D) The Fibronectin matrix mediates inter-tissue adhesion like a 'smart glue' that dynamically remodels in a graded fashion at the PSM|NT interface in response to neural tube convergence to have higher density at zones of higher inter-tissue stress. This inter-tissue adhesion mechanically couples left and right PSM to the neural tube like a 'lap joint' and maintain bilateral symmetry during elongation. The Fibronectin matrix also provides resistance (green arrows) against the neural tube convergence (black arrow). Thus, the ECM can act like a double-edged sword. Too much matrix deposition maintains the symmetry, but slows down the convergence due to high resistance, and predisposes the neural tube to spina bifida-like phenotype with an open neural tube (*cdh2* mutant, left panel). Conversely, reduction in matrix deposition helps the neural tube convergence by reducing the resistance (as shown by the rescue of neural tube convergence in double *cdh2; itgα5* mutants), but it produces local tissue detachment and breaks the mechanical coupling between the tissues generating left-right asymmetries (*itgα5* mutants, right panel).

Fibronectin matrix, F-actin and traction forces. The combination of Fibronectin's function as a mediator of mechanical forces, a stress sensor, and positive feedback with actomyosin contractility make Fibronectin a smart adhesive. This smart adhesive activity is revealed in our Fibronectin photoconversion experiments which show that the Fibronectin matrix continually remodels in response to neural tube convergence to create the medial-lateral gradient of matrix.

Folate deficiency is the best-known cause of spina bifida in humans, but it is currently unclear how this deficiency leads to a failure of neural tube convergence and closure. Genetic causes of spina bifida include mutations in the planar cell polarity pathway, which directs convergent-extension, and mutations in genes that regulate cytoskeletal remodeling during apical constriction. These two groups of mutations have more direct effects on neural tube morphogenesis, and thus the etiologies of these defects seem clear (*Wallingford et al., 2013*; *Copp et al., 2015*). Cadherin-2-mediated cell-cell adhesion is required for neural tube morphogenesis, but here we find that *cadherin 2* mutant neural tubes are capable of convergence as long as they do not have the additional work of coupling the left and right paraxial mesoderm. Thus, the neural tube has to be maximally fit in order to converge and fuse, and any environmental or genetic deficiency that weakens the neural tube may lead to spina bifida. Indeed, in humans, neural differentiation appears to be relatively normal during failure of neural tube closure, but exposure to the amniotic fluid, which becomes toxic later in gestation, causes neural degeneration. This is likely a reason why prenatal surgical repair limits the neurological consequences of spina bifida: in the absence of degeneration more normal neural function can be achieved. Thus, folate deficiency may directly affect cell proliferation or epigenetics without dramatically altering neural differentiation (*Copp et al., 2015*). Folate increases cell traction force in neuronal cell culture (*Kim et al., 2018*), thus folate deficiency may cause spina bifida by diminishing the overall mechanical fitness of the neural tube. In principle, the mechanical consequences of folate deficiency on the neural tube could be tested in an animal model. Moreover, mobilization from the flanking mesoderm could rescue spinal cord closure in folate deficient and other environmentally and genetically weakened neural tubes.

Based on systematic analysis of cell motion in the tailbud, we previously hypothesized that PSM tissue assembly may involve a fluid to solid transition (*McMillen and Holley, 2015*). Recent experiments identifying gradual increases in tissue stiffness and cell packing during PSM maturation support this idea (*Mongera et al., 2018*). Our data here extend this model as we find that there are posterior to anterior increases in (1) F-actin around the cell cortices of PSM cells, (2) epithelization of cells on the surface of the PSM and (3) assembly of large Fibronectin fibers. This posterior to anterior pattern is consistent with the established relationship between cortical tension, Fibronectin matrix assembly and traction forces. Notably, the anterior-posterior patterns of Fibronectin and F-actin are only observed on the lateral surface of the PSM which interfaces the epidermis. Medially, along the neural tube-PSM interface, the lap joint mechanics predominate, and medial to lateral gradients of F-actin and Fibronectin are observed.

While our analysis has focused on the development of the posterior trunk, our model likely applies to the more anterior neural tube because it displays even more convergence. However, there is less convergence during the formation of the more posterior spinal cord in the tail. Thus, ECM-mediated inter-tissue adhesion is likely less of a mechanical impediment to the development of the most caudal spinal cord. However, during these later stages of body elongation, inter-tissue adhesion is important for establishing the chevron shape of the myotome which is the major derivative of the zebrafish somite (*Tlili et al., 2019*).

In summary, we find that the Fibronectin matrix ensures symmetric morphogenesis of the spinal column. During development, mechanical forces among attached tissues are intrinsically coupled to the ever-changing geometry, and to preserve symmetries, these forces need to be actively coordinated. In the course of vertebrate body elongation, we find that tissue mechanics are tuned to accomplish the competing goals of neural tube convergence and maintenance of bilateral symmetry. The mechanism that dynamically couples these embryonic tissues resembles adhesive lap joints which are utilized in processes ranging from aerospace manufacturing to woodworking. Thus, microscale embryo biomechanics can function very much like macroscale engineered structures.

# Materials and methods

## Key resources table

| Reagent type (species) or resource | Designation | Source or reference | Identifiers | Additional information |
|---|---|---|---|---|
| Strain, strain background (*Danio rerio* male and female) | Wild-type strain TLAB | ZIRC | | Crosses from AB strain (RRID:ZIRC_ZL1) with TL strain (RRID:ZIRC_ZL86) |
| Strain, strain background (*Danio rerio* male and female) | Wild-type strain TLF | ZIRC | | RRID:ZIRC_ZL86 |
| Genetic reagent (*Danio rerio*) | strain cdh2 mutant tm101 | *Lele et al., 2002* | | RRID: ZFIN_ZDB-GENO-080110-3 |
| Genetic reagent (*Danio rerio*) | strain MZ itgα5 mutant thl30 | *Jülich et al., 2005* | | ZIRC ID: ZL2023 |
| Genetic reagent (*Danio rerio*) | Tg(*hsp70:fn1a-mKIKGR*) | This paper | | Transgenic line expressing, , fibronectin 1a tagged with mKIKGR under the control of the heat-shock promoter hsp70. See Material and methods section. Available in Scott Holley laboratory (Yale University) |
| Genetic reagent (*Danio rerio*) | fn1a; fn1b double mutant | This paper | | CRISPR/Cas9 generated mutant line where fn1a and fn1b genes have been knocked out by insertion of a stop cassette. See Material and methods section. Available in Scott Holley laboratory (Yale University) |
| Antibody | Anti-Fibronectin antibody (Rabbit polyclonal) | SIGMA | F3648 | 1/100 RRID:AB_476976 |
| Antibody | Anti-V5 antibody (Goat polyclonal) | Abcam | Ab 9137 | 1/400 RRID:AB_307037 |
| Antibody | H5-V5-tag antibody (recombinant purified scFv) | *Cao et al., 2017* | | 50 mg/ml |
| Antibody | Anti-goat Alexa 555 (Donkey polyclonal) | Thermofisher scientific | A32816 | 1/200 RRID:AB_2762839 |
| Antibody | Anti-rabbit Alexa 555 (Donkey polyclonal) | Invitrogen | A31572 | 1/200 RRID:AB_162543 |
| Sequence-based reagent | sgRNA fn1a-/-line | This paper | | 5'ATTTAGGTGACACTATAGGAGGGCACTCC TACAAGATGTTTTAGAGCTAGAAATAGCAAG3' |
| Sequence-based reagent | stop codon cassette oligonucleotide fn1a-/-line | This paper | | 5'GAGGGAGGGCACTCCTACAAGTCATGGCGTT TAAACCTTAATTAAGCTGTTGTAGGATT GGAGACACATGGCAGA3' |
| Sequence-based reagent | sgRNA fn1b-/-line | This paper | | 5'ATTTAGGTGACACTATAGGACTGCACATGTTT GGGAGGTTTTAGAGCTAGAAATAGCAAG3' |
| Sequence-based reagent | stop codon cassette oligonucleotide fn1b-/-line | This paper | | 5'CGTGGACTGCACATGTTTGGGTCATGGCGTTTAAA CCTTAATTAAGCTGTTGTAGGAGAGGGAAACGGACGCATC3' |
| Sequence-based reagent | fn1a DiagA1 primer | This paper | | 5'GACTGTACTTGCATTGGCTCTG3' |

*Continued on next page*

*Continued*

| Reagent type (species) or resource | Designation | Source or reference | Identifiers | Additional information |
|---|---|---|---|---|
| Sequence-based reagent | fn1b DiagA1 primer | This paper | | 5'GAGCGTTGCTATGATGACTCAC3' |
| Sequence-based reagent | stop A primer | This paper | | 5'GCTTAATTAAGGTTTAAACGCC3' |
| Recombinant DNA reagent | mKikGR plasmid | *Habuchi et al., 2008* | | |
| Recombinant DNA reagent | Tg(*hsp70:fn1a-mKIKGR*) plasmid | This paper | | Plasmid designed to generate the Tg(*hsp70:fn1a-mKIKGR*) transgenic line. See Materials and methods section for detailed information about the sequence of this transgene. Available in Scott Holley laboratory (Yale University) |
| Recombinant DNA reagent | FN1a-mKIKGR13.2-hsFNIII10-11 plasmid | This paper | | Plasmid designed to generate the mRNA coding for the human/zebrafish chimeric FN1a-mKIKGR13.2-hsFNIII10-11 protein. See Materials and methods section. Available in Scott Holley laboratory (Yale University) |
| Software, algorithm | Imaris | Bitplane | | RRID:SCR_007370 |
| Software, algorithm | Matlab | Mathworks | | RRID:SCR_001622 |
| Software, algorithm | Fiji | Opensource | | RRID:SCR_002285 |
| Other | Alexa Fluor 488 Phalloidin | Life technologies | A12379 | |

## Animal models

Zebrafish were raised according to standard protocols (*Nüsslein-Volhard and Dahm, 2002*) and approved by the Yale Institutional Animal Care and Use Committee. The TLAB and TLF wild-type strain were used. The tm101B allele of *cdh2* was used (*Lele et al., 2002*). The MZ itgα5$^{-/-}$ mutant line is a maternal zygotic mutant line using the thl30 allele (*Jülich et al., 2005*). The single mutant lines and double mutant lines were crossed to the *Tg (hsp70:fn1a-mKIKGR)* line to the generate the mutants lines in transgenic background used in photoconversion experiments. Embryos were collected from pair-wise natural matings. Sex-specific data were not collected as zebrafish do not have a strictly genetic sex determination mechanism, and sex is determined after the first 36 hr of development studied here (*Wilson et al., 2014*).

## Generation of a double mutant line for fn1a and fn1b genes

To generate a double mutant line for *fn1a* and *fn1b* genes, we first created single mutant lines for *fn1a* and *fn1b* genes separately using CRISPR. Single cell stage embryos were injected with a mix containing: Cas9 mRNA, a single guide RNA (sgRNA) specifically designed to target a chosen sequence in the exon 4 of *fn1a* gene or in the exon 5 of *fn1b* gene (see *Figure 5—figure supplement 1* for detailed target sequences, and a stop codon cassette oligonucleotide *Gagnon et al., 2014*). Sequences for the *fn1a$^{-/-}$* line: sgRNA(5'ATTTAGGTGACACTATAGGAGGGCACTCCTACAAGATGTTTTAGAGCTAGAAATAGCAAG3'); stop codon cassette oligonucleotide (5'GAGGGAGGGCACTCCTACAAGTCATGGCGTTTAAACCTTAATTAAGCTGTTGTAGGATTGGAGACACATGGCAGA3'). Sequences for the *fn1b$^{-/-}$* line: sgRNA(5'ATTTAGGTGACACTATAGGACTGCACATGTTTGGGAGGTTTTAGAGCTAGAAATAGCAAG3'); stop codon cassette oligonucleotide (5'CGTGGACTGCACATGTTTGGGTCATGGCGTTTAAACCTTAATTAAGCTGTTGTAGGAGAGGGAAACGGACGCATC3').

General details regarding the choice of target sequence, the design of the stop codon cassette oligonucleotide, the design and synthesis of the sgRNA and conditions of injection can be found in the detailed supplemental protocol described in *Gagnon et al. (2014)*. Individuals positive for the insertion of the stop codon cassette were identified by PCR on genomic DNA using *fn1a* DiagA1/ stopA primers or *fn1b* DiagA1/stopA primers: *fn1a* Diag A1(5'GACTGTACTTGCATTGGCTCTG3') *fn1b* DiagA1 (5'GAGCGTTGCTATGATGACTCAC3') stop A (5'GCTTAATTAAGGTTTAAACGCC3').

Then single mutant lines were then crossed together to obtain the double mutant line.

## *Tg (hsp70:fn1a-mKIKGR)* transgenic line generation and heat-shock driven transgene expression

Information concerning the sequence of the transgene is described in *Figure 5—figure supplement 1A*. The plasmid construct containing the transgene was generated following the previously described method (*Jülich et al., 2015*) with the mKikGR plasmid provided by the Atsushi Miyawaki laboratory (*Habuchi et al., 2008*). 75 ng/µl of plasmid were co-injected with 120 ng/µl of Tol2 mRNA at 1 cell stage. For both in vivo tracking of Fibronectin matrix dynamics and rescue experiments, embryos were heat-shocked by incubation 30 min at 38°C.

## Fibronectin immunostaining, and F-actin and nuclei labeling

12–14 somite stage embryos were fixed overnight (4°C) in 4% PFA (in 1X PBS). After dechorionation in 1X PBS, embryos were rinsed 3 × 5 min in PBSDT (1% DMSO, 0.1% Triton in 1X PBS), permeabilized 3 min with Proteinase K (5 µg/ml in PBSDT), rinsed 2 × 5 min in PBSDT, post-fixed 20 min at RT in 4% PFA and finally rinsed 3 × 5 min in PBSDT. Blocking was done 2–3 hr at RT in blocking solution (1% Blocking reagent from Roche in PBSDT). Embryos were incubated O/N at 4°C with the primary antibody against Fibronectin (rabbit polyclonal SIGMA, F3648) (1/100 dilution in blocking solution). Embryos were rinsed 2 × 15 min in blocking solution, 2 × 15 min and 3 × 1 hr in PBSDT. Embryos were incubated O/N at 4°C with both secondary antibody (Alexa fluor 555 donkey anti-rabbit IgG, Invitrogen A31572) and Alexa Fluor 488 Phalloidin (Life technologies, A12379) respectively diluted 1/200 and 1/100 in blocking solution. Embryos were incubated 15 min with DAPI solution (100 pg/ml in PBSDT), and then rinsed 2 × 15 min and 3 × 1 hr in PBSDT.

## Confocal and lightsheet imaging

For morphometric analysis, whole embryos were mounted in glass capillaries (1.4 mm diameter) to preserve tissue morphologies (1% agarose in 1X PBS) and imaged with a Lightsheet Z.1 (ZEISS, 20x objective). For imaging of fibronectin matrix and F-actin at high resolution, only the tails of the embryos were mounted between standard slides and coverslips (Fisher Finest Premium Microscope Slides) in 50% glycerol in 1X PBS and imaged with an inverted confocal microscope (ZEISS LSM880, 40x oil objective). For the photoconversion assay, 12 s stage embryos were mounted in glass bottom dishes in drops of 1% agarose in 1X E2 covered with E2 medium. An inverted confocal microscope ZEISS LSM880, 20x objective) was used for both photoconversion and subsequent imaging of the photoconverted region. Regions of interest were photoconverted using a 405 nm laser (30–45 cycles, speed 9 average 1, 10% of maximum laser power).

## Simulation methods

We model the 2D cross-section of the posterior tail (see *Figure 2A* in main text) based on an earlier model of soft grains (*Åström and Karttunen, 2006*). The tissue surfaces are modeled by closed loops of mass-spring chains. An isolated tissue cross-section is made of 50 mass-points connected by elastic springs with homogeneous stiffness constant $K_S$. This parameter ($K_S$) is named 'surface stiffness'. Thus, the tension force along the $i$-th spring is: $\vec{T}_i = K_S(l_i - l_0)h_i$, where $l_i$ and $l_0$ are the instantaneous length and the equilibrium (unstretched) length of the $i$-th spring respectively, and $h_i$ is a unit vector along the $i$-th spring. The sum of these tension forces along all the surface springs represents the surface tension of the tissue. An internal pressure, $P$ also acts in the bulk of each tissue and it is directed perpendicularly outward at the tissue surface. Therefore, the net force acting on the $i$-th mass point of an isolated tissue ($\vec{F}_i^{tissue}$) is given by:

$$\overrightarrow{F}_i^{\,tissue} = \overrightarrow{T}_i - \overrightarrow{T}_{i+1} + \frac{P\,l_0}{2}(n_i + n_{i+1})$$

Here, $n_i$ denotes an outward directed unit vector perpendicular to the $i$-th spring. We next modeled the interaction forces between adjacent tissues. Two mass-points belonging to two different tissues are considered to be interacting only if they are within a distance $R_{adh}$ from each other. If this distance between the mass-points is between $R_{rep}$ and $R_{adh}$, there exists a spring-like adhesive force that represents 'inter-tissue adhesion'. In addition, to prevent tissues from penetrating into each other, we implemented 'volume exclusion' by assuming spring-like repulsion forces between the mass-points, when the distance between the mass-points is below $R_{rep}$. Therefore, the interaction forces ($\overrightarrow{F}_{ij}^{\,int}$) between $i$-th and $j$-th mass-points belonging to adjacent tissues can be summarized as:

$$\overrightarrow{F}_{ij}^{\,int} = K_{rep}\left(R_{rep} - r_{ij}\right)\hat{r}_{ij}, if\, r_{ij} < R_{rep}$$
$$or, \overrightarrow{F}_{ij}^{\,int} = -K_{adh}\left(r_{ij} - R_{rep}\right)\hat{r}_{ij}, if\, R_{rep} \leq r_{ij} \leq R_{adh}$$
$$or, \overrightarrow{F}_{ij}^{\,int} = 0, if\, r_{ij} > R_{adh}$$

Here, $r_{ij}$ is the metric distance between the mass-points (i.e. $r_{ij} = |\overrightarrow{r}_i - \overrightarrow{r}_j|$) and $r_{ij}$ is a unit vector pointing from the $j$-th to the $i$-th point. $K_{adh}$ and $K_{rep}$ represent the strength of adhesive and repulsive interactions respectively. In general, $K_{rep}$ is much higher than $K_{adh}$, and it is kept constant in all simulations. On the hand, a variation in $K_{adh}$ represents a variation in the level of inter-tissue adhesion. This parameter, $K_{adh}$ is referred as 'adhesion stiffness' that represents the strength of cell-ECM interaction.

We finally assumed an over-damped dynamics (neglecting inertia) of the mass-points to evolve their positions over time as below:

$$c\,\overrightarrow{v}_i = \overrightarrow{F}_i^{\,tissue} + \sum_j \overrightarrow{F}_{ij}^{\,int}$$

Here, $\overrightarrow{v}_i$ is the velocity of $i$-th mass-point and $c$ is the viscous coefficient representing a drag force. By evolving the above dynamics in computer, we produced steady final shapes of the tissues at the limit of very long time. We used the 'Explicit Euler' method to simulate the dynamics with a time step set at $\Delta t = 0.001$.

## Parameter values

In all simulations, the equilibrium spring lengths ($l_0$), the repulsion strength ($K_{rep}$), the viscous coefficient ($c$), the range of forces between tissues ($R_{rep}$ and $R_{adh}$) and the internal pressure ($P$) are kept constant. These fixed parameters are: $l_0 = 0.1, K_{rep} = 30000, c = 10, R_{rep} = 0.1, R_{adh} = 0.2$, and $P = 5$. To theoretically explore the impact of variations of the surface stiffness ($K_s$) and adhesion stiffness ($K_{adh}$) in *Figure 2B*, we used $K_{adh} = 4$ and $K_{adh} = 12$, while keeping the surface stiffness fixed at $K_s = 50$. We also used $K_s = 35$ and $K_s = 100$, keeping the adhesion stiffness fixed at $K_{adh} = 8$. We next systematically varied the surface stiffness ($K_s$) and adhesion stiffness ($K_{adh}$) to mimic the conditions of different genotypes in *Figure 2D, E and F*. The chosen values for these parameters are: for wild-type $K_s = 100$, $K_{adh} = 10$; for *cdh2⁻ᐟ⁻* mutant $K_s = 55$, $K_{adh} = 12$; for *itgα5⁻ᐟ⁻* mutants $K_s = 100$, $K_{adh} = 6.5$; and for *cdh2⁻ᐟ⁻;itgα5⁻ᐟ⁻* mutants $K_s = 55$, $K_{adh} = 6.5$ (also see the parameter space plot in *Figure 2D*). In *Figures 3A* and *7A*, we used $K_s = 200$ and $K_{adh} = 10$ to generate the symmetric tension distributions along the PSM|NT interfaces.

## Morphometric analysis

By using Imaris, we made transverse sections of 3D reconstructs of the posterior tail in every 20 um starting from the last somite boundary. Each transverse section was exported into image-J and rotated such that the medial-lateral axis of the notochord is exactly parallel to the horizontal axis of the image. We then traced the contours of neural tube and PSM to define regions of interests (ROIs). We applied the 'bounding rectangle' tool to these ROIs and the medial-lateral and dorsal-ventral lengths of the tissues are given by the width and heights of the rectangle. The same ROIs

were also used to determine the cross-sectional area of the tissues (Path in image-J: Analyze - > Set measurements - > select 'area' and 'bounding rectangle'). The interfacial length and angle were measured with standard 'segmented line tool' and 'angle tool' from image J. The radius of curvature was measured using a online 'Fit circle' Image J macro adapted from Pratt V., 1987 (see the website: http://www.math.uab.edu/~chernov/cl/MATLABcircle.html, http://imagej.1557.x6.nabble.com/Re-bending-and-radius-of-curvature-td3686117.html). We provide the image-J macro at the end of the methods section.

## Quantitative analysis of matrix assembly and F-actin signal

### Image processing

From 3D reconstructs of the posterior tail, portions of the tissue interfaces were isolated and oriented in Imaris such that the view is perpendicular to the interface. They were then stitched together using Photoshop to reconstruct the full interface.

For the analysis of matrix assembly, each image of the full interface was individually adjusted for brightness and contrast and then thresholded with an 'auto-local threshold' in image J to capture most of the matrix content (Image J path: Image - > Adjust - > auto local threshold - > select method 'Pahnsalkar', radius = 15). After thresholding images were then transformed into binaries. Based on the particle size in these binarized images, we considered the elements below 4 µm as background noise and removed them. After noise removal, what remains are referred as 'total matrix' (elements above 4 µm). We further made two categories of matrix elements: small fibrils (4 µm −40 µm), and large assembled networks (greater than 40 µm).

For the analysis of F-actin signals, each image of the full interface was converted into a binary using the matlab 'im2bw' function and with a global threshold set at 20% above the minimum intensity for every sample.

### Quantification from processed images

In the PSM|NT interface, we measured the medial-lateral (ML) distribution of binary signals representing either one of the matrix categories or F-actin. We sectioned the interface along the anterior-posterior axis in consecutive sectors of roughly 10 µm in width. In each section, the relative ML positions of the pixels with respect to the local position of notochord were listed. From this list, we built a histogram of pixel counts as a function of relative ML position, normalized by the total pixel count of that interface.

In the PSM|E interface, we measured the density of binary signals representing either one of the matrix categories or F-actin along the anterior-posterior axis. To measure the density, a binary image representing total tissue surface was also prepared by tracing the contours of the total interface and filling the enclosed region with white pixels and outside with black. We then sectioned the interface along the anterior-posterior axis in consecutive sectors of roughly 50 µm in width. In each section, the density was measured by dividing the total pixel count of matrix or F-actin by the total pixels present in the local tissue area. These quantifications were performed by custom Matlab codes provided below:

Medial-lateral distribution of F-actin or matrix elements:

```
IMECMassemle = imread('TOTALMATRIX.tif');
IMECMassemle = im2bw(IMECMassemle);
IMECMunassemle = imread('UNASSEMBLEDMATRIX.tif');
IMECMunassemle = im2bw(IMECMunassemle);
IMECMtot = imread('TOTALTISSUE.tif');
IMECMtot = im2bw(IMECMtot);
[x1,y1]=size(IMECMassemle);
height = min(x1,y1);
width = max(x1,y1);
dx = 10*17.4; %round(width/Ntot);
Ntot = round(width/dx);
%k = 0;
xmin = 0;
```

```
%Distmin = 25;
for k = 1:Ntot
filename = strcat('Icrop', num2str(k), '.tif');
jjassemble = imcrop(IMECMassemle,[xmin,0,dx,height]);
jjunassemble = imcrop(IMECMunassemle,[xmin,0,dx,height]);
jjtot = imcrop(IMECMtot,[xmin,0,dx,height]);
%imwrite(jj,filename)
[rowsas,colsas]=find(jjassemble == 0); %%%% cols = AP, rows = ML, NT position = max
(rows)
[rowsunas,colsunas] =find(jjunassemble == 0);
[rowstot,colstot]=find(jjtot == 0);
NTposition = max(rowstot);
ML = max(rowstot)-min(rowstot);
rowsas1{k}=abs(rowsas-NTposition)/ML;
rowsunas1{k}=abs(rowsunas-NTposition)/ML;
xmin = xmin + dx;
%Dist(k)=Distmin;
%Distmin = Distmin+50;
%%dist = 20 um
end
MLas = cell2mat(rowsas1');
MLunas = cell2mat(rowsunas1');
hist(MLas,30); hold on
hist(MLunas,30);
[fas,xas]=hist(MLas,30);
[funas,xunas]=hist(MLunas,30);
%output=[mode(MLas), mode(MLunas), mean(MLas), mean(MLunas), std(MLas), std
(MLunas)];
%xlswrite('Spatial-Segregation_WT-1_L1-PSML_6-22-17.xls',output)
output=[xas',fas',xunas',funas'];
%xlswrite('ML_Frequency_WT-1_L1-PSML_6-22-17.xls',output)
%save ML_Frequency_WT-1_L1-PSML_6-22-17.dat output -ascii
Anterior-posterior density of F-actin or matrix elements:
ImageECM = imread('ASSEMBLEDMATRIX.tif');
BWECM = im2bw(ImageECM);
nBlack = sum(BWECM(:)==0);
[x1,y1]=size(BWECM); %%%%%% check if size is same for ALL FIGURES
height = min(x1,y1);
width = max(x1,y1);
dx = 50*17.4; %round(width/Ntot);
Ntot = round(width/dx);
k = 0;
xmin = 0;
Distmin = 25;
for k = 1:Ntot
filename = strcat('Icrop', num2str(k), '.tif');
jj = imcrop(BWECM,[xmin,0,dx,height]);
imwrite(jj,filename)
xmin = xmin + dx;
Dist(k)=Distmin;
Distmin = Distmin+50;
%%dist = 20 um
end
for k = 1:Ntot
filename = strcat('Icrop', num2str(k), '.tif');
IMcrop = imread(filename);
```

```
%BWcrop = im2bw(IMcrop);
nB(k)=sum(IMcrop(:)==0);
end
%Dist = 1:Ntot;
output=[Dist',nB'];
xlswrite('assembled.xls',output)
```

## Quantification of cell aspect ratio and F-actin intensity in PSM

To get the cell aspect ratio, we first prepared cross-sections of the PSM prepared by the Imaris software. In these cross-sections, each cell was manually traced and aspect ratio (major axis/minor axis) was extracted using the 'Shape descriptor' plugin of ImageJ (path: analyze - > Set measurements - > click 'shape descriptor').

To quantify the mean F-actin intensity of the PSM surface cells or PSM internal cells, we first traced a region of interest (ROI) encompassing either surface or internal cells. Then the mean grey value within the ROI was extracted after setting a threshold. The threshold was set such that we remove the cytoplasmic background without removing the low membrane signals in the posterior of the PSM.

## Quantitative analysis of matrix remodeling

We first oriented the images of photoconverted spot in Imaris to get a view perpendicular to the spot in each time frame, and cropped around the spot. These images were then manually thresholded by Image J (path: Image - > Adjust - > Threshold) to account for photobleaching and then converted into binary images. From these binarized images, we used the standard deviation of the white pixel distribution along the medial-lateral and anterior-posterior axes as measures of the medial-lateral width and anterior-posterior height of the photoconverted spot respectively. These metrics were measured by a custom Matlab code provided below:

```
for k = 1:8
filename = strcat('SLICE', num2str(k), '.tif');
IMECM = imread(filename);
IMECM = im2bw(IMECM);
nwhite = nnz(IMECM);
[rows,cols]=find(IMECM == 1);
Time(k)=(k-1)*15;
area(k)=nwhite*0.2*0.2;
width(k)=std(cols); %./mean(cols);
height(k)=std(rows); %./mean(rows);
%aspect(k)=allaspect(biggest);
%NoElement = CC.NumObjects
end
output=[Time', width', height', width'/width(1), height'/height(1)];
plot(Time',height'/height(1),'o k-'); hold on
plot(Time', width'/width(1),'o r-')
xlswrite('RedSpot_WT-4_PSML.xls',output)
```

## Displacement field analysis of the photoconverted region

To determine if there is any directional bias in the medial-lateral remodeling of matrix at the PSM|NT interface, we analyzed the displacement field of the photoconverted signal using a Particle Image Velocimetry (PIV) plugin available in Image J (downloadable versions and tutorials at https://sites.google.com/site/qingzongtseng/piv#tuto). For this analysis, images of photoconverted spot were prepared in Imaris to get a view perpendicular to the spot in each time frame. We used the raw grey scale images with full intensity distribution (without any thresholding). In Image-J, we selected the 'iterative PIV (advanced)' method, which determines the correlation between two

images at a time and requires three user-defined parameters: the 'interrogation window size' (IW), the 'search window size' (SW), and the 'vector spacing' (VS). The set of two images is analyzed by PIV through three passes with a progressive decrease in parameter values so that a fine-tuned vector field of pixel displacement is finally produced. For our analysis, the set of two images corresponds to two consecutive time frames of the spot and we chose the following parameter values: for 1st pass, IW = 60, SW = 120, VS = 15; for 2nd pass, IW = 50, SW = 120, VS = 12; and for 3rd pass, IW = 40, SW = 80, VS = 10. The displacement was analyzed over the course of 45 min (using a frame to frame correlation for the first four consecutive time frames) as most of the remodeling of the photoconverted region happened in the first hour. All the vectors from the Frame 1–2, Frame 2–3 and Frame 3–4 correlations were plotted together in the same rose plot (see *Figure 6—figure supplement 1E*).

## Quantitative analysis of myl12.1-eGFP localization in live embryos

mRNA coding for myl12.1-eGFP was synthesized using mMessage mMachine SP6 kit (Thermofisher, AM1340) according to standard manufacturer instructions using a PCS2+ myl12.1-eGFP plasmid as a template (a gift from Carl-Philipp Heisenberg). Embryos were injected at the one-cell stage with 100 pg of myl12.1-eGFP mRNA (25 ng/µl with a 4 nl droplet). Live embryos were imaged at the 12–14 somites stage on a ZEISS LSM880 (40x water objective, zoom 2, with airyscan detector set in a fast mode acquisition). Each image stack was then automatically sliced transversally every 5 µm using the 'Reslice tool' in Fiji. To quantify medio-lateral differences in myl12.1-eGFP localization along the NT border on each transverse slice, we used a custom Fiji code (see code below) to divide the NT border into 3 regions of 3–5 µm in width and measured the mean intensity value in each region.

```
roiManager("Select", newArray(0,1,6,7));
roiManager("Combine");
getSelectionCoordinates(xpoints, ypoints);
makeSelection("polygon", xpoints, ypoints);
run("Measure");
roiManager("Deselect");
roiManager("Select", newArray(1,2,5,6));
roiManager("Combine");
getSelectionCoordinates(xpoints, ypoints);
makeSelection("polygon", xpoints, ypoints);
run("Measure");
roiManager("Deselect");
roiManager("Select", newArray(2,3,4,5));
roiManager("Combine");
getSelectionCoordinates(xpoints, ypoints);
makeSelection("polygon", xpoints, ypoints);
run("Measure");
roiManager("Deselect");
roiManager("Select", newArray(2,3,4,5));
roiManager("Combine");
getSelectionCoordinates(xpoints, ypoints);
makeSelection("polygon", xpoints, ypoints);
run("Measure");
roiManager("Deselect");
roiManager("Save", "/image path/ROIn.zip");
roiManager("Deselect");
roiManager("Delete");
selectWindow("Results");
String.copyResults();
Table.deleteRows(0, 2);
```

## Quantification of tension within the fibronectin matrix via the H5 antibody

To analyze the tension within the Fibronectin matrix, we used the H5 monoclonal antibody which recognizes a tension dependent epitope on the human Fibronectin 10th FN type III repeat (*Cao et al., 2017*). We generated a chimeric FN1a-mKIKGR13.2-hsFNIII10-11 protein in which the zebrafish 10th and 11th FN type III repeats have been replaced by the human ortholog's 10th and 11th FN type III repeats. (The 10th FN type III repeat contains the PSHRN motif and was formerly identified as the 9th FN type III repeat. The 11th FN type III repeat contains the RGD motif and was formerly identified as the 10th FN type III repeat.) The PCS2+ FN1a-mKIKGR13.2-hsFNIII10-11 construct was generated by first subcloning the FN1a-mKIKGR13.2 CDS into the PCS2+ plasmid. The zebrafish FNIII10-11 repeats were replaced with a zebrafish codon optimized CDS of the human ortholog FNIII10-11 repeats using Gibson assembly. To synthetize the mRNA, we followed the protocol of *Prince and Jessen (2019)*. Briefly, 30 μg of plasmid was linearized with Not I-HF (200 μl total volume reaction with 8 μl of enzyme) and purified via phenol chloroform extraction, overnight sodium acetate/ethanol precipitation and eluted in 10 μl RNAse free water. mRNA was synthesized using the mMessage Machine SP6 kit (Thermofisher, AM1340) following manufacturer's instructions for long mRNA synthesis (1 μl of GTP was added to the reaction mix (total volume 21 μl) with 1 μg of DNA template at high concentration (1–3 μl of DNA in the final reaction mix).

One-cell stage embryos were injected with 600–800 pg of FN1a-mKIKGR13.2-hsFNIII10-11 mRNA. Embryos were sorted for bright fluorescence at the 12–14 somite stage, fixed 40 min in 4% PFA (gentle lateral shaking) at RT, washed twice for 5 min in PBST (0.1% Tween in 1X PBS), dechorionated, washed twice for 5 min in PBST, treated for 3 min with 5 μg/ml Proteinase K diluted in PBST, washed 3 × 5 min in PBST, post-fixed 20 min in 4% PFA at RT(gentle lateral shaking), washed 2 × 5 min in PBST, blocked for 3.5 hr at RT (gentle lateral shaking) in blocking solution (2% blocking reagent from Roche, 1X maleic acid buffer (2X maleic acid buffer: 200 mM maleic acid 300 mM NaCl pH = 7.4)), incubated 20 hr at 4°C with H5-V5-tag antibody (a gift from Thomas Barker) diluted at 50 μg/ml in blocking solution, washed 4 × 15 min in 1X maleic acid buffer (gentle lateral shaking at RT), incubated 20H at 4°C with goat anti-V5 antibody (Abcam, Ab 9137 diluted 1/400) in blocking solution, washed 4 × 15 min in 1X maleic acid buffer (gentle lateral shaking at RT), incubated 20H at 4°C with donkey anti-goat Alexa 555 antibodies (Thermofisher scientific A32816) diluted 1/200 in blocking solution, washed 3 × 10 min in 1X maleic acid buffer (gentle lateral shaking at RT), checked for fluorescence, incubated in 25% and 50% glycerol (10 min each) and then mounted in 50% glycerol for confocal imaging (Zeiss LSM880, 40x water objective). All antibodies incubations were done with 50 μl antibody solution without shaking.

To quantify H5 and mKIKGR levels and the H5/mKIKGR ratio, H5 and mKIKGR signals were imaged in conditions minimizing pixel saturation. Tissue interfaces were then segmented using Imaris software and exported for analysis with display adjustments preventing any distortion of the raw pixel values (gamma 1, full dynamic range 0–255). From this step onward, further image treatment and quantification were performed using a custom MATLAB code (see code provided below). To summarize, the H5 and mKIK images were individually thresholded (based on visual assessment) to define background pixels corresponding to areas between matrix fibrils and to discard any saturated pixels (value = 255, which represented than 0.8% of pixels per image, on average). We then generated the H5/mKIK ratiometric image by dividing, pixel by pixel, the H5 image by the mKIKGR image, and displayed it with a scaled color based on pixel value (imagesc MATLAB function). For the quantification of the medial-lateral distributions of each signal, images were divided in 10 sections of equal size along the medial-lateral axis, and the average signal in each section was quantified and then normalized to the average signal in the most lateral section. For the H5 and mKIKGR levels analysis, the calculation of the average signal in each section included the background pixels, which were attributed a value of zero. Thus, the tissue scale average incorporates both matrix brightness as well as matrix density. By contrast, for the H5/mKIKGR ratio analysis, the calculation of the average signal in each section excluded the background pixels that were attributed a NaN value. Thus, this metric reflects the average signal within the matrix elements themselves and represents the local level of tension within the Fibronectin matrix.

MATLAB code for quantification of medial-lateral distribution of H5/mKIK ratio and H5, mKIK signals

```
clr = 'rgbk';
lw = 3;
fs = 20;
th1 = 33;
th2 = 30;
% th1 and th2 are manually determined to exclude zones in between matrix elements
imgs = {'H5.tif','mKIK.tif','ratio.tif'};
img1 = double(imread(imgs{1}));
img2 = double(imread(imgs{2}));
img1_255 = img1(img1 == 255);
img2_255 = img2(img2 == 255);
percent255_1 = numel(img1_255)/numel(img1)
percent255_2 = numel(img2_255)/numel(img2)
img1(img1 == 0)=NaN;
img2(img2 == 0)=NaN;
% 2 previous lines exclude the zones of picture outside of the tissue
img1(img1 == 255)=NaN;
img2(img2 == 255)=NaN;
% 2 previous lines exclude the saturated pixels
img1(img1 <th1)=NaN;
img2(img2 <th2)=NaN;
% Values in the 2 previous lines are NaN for an averaging calulating the brightness
of the matrix
% content itself. Change them to 0 for an avering taking into account the
% total surface of the tissue
imgRatio = (img1./img2);
imgRatio(imgRatio == Inf)=NaN;
imgRatio(imgRatio == 0)=NaN;
fig = figure('Renderer', 'painters', 'Position', [0 0 size(imgRatio,2) size
(imgRatio,1)]);
imagesc(imgRatio);
caxis([0 4]); %change the upper value to tune the colorbar and colormap
colorbar;
box off;
axis off;
saveas(fig,'ratio.tif','tif');
nbin = 10;
img = {img1, img2, imgRatio};
fig = figure;
figure(2);
hold on;
for i = 1:size(imgs,2)
[imgmean, imgstd, imgmeannorm, position,scale]=mlGradient(img{i}, nbin);
plot(position, imgmeannorm, clr(i), 'LineWidth', lw, 'DisplayName', imgs{i});
set(gca, 'xminorgrid', 'off', 'yminorgrid', 'off', 'fontsize', fs, 'xminortick',
'off',...
'yminortick', 'off', 'linewidth', lw);
set(gcf, 'color', 'w');
legend('Location','bestoutside','FontWeight','Normal','FontSize',12);
set(gca,'FontSize',fs,'linewidth',lw);
box on;
ylabel('Average pixel intensity');
xlabel('Medio-lateral position (relative)');
end
Function mlGradient
function [imgmean,imgstd,imgmeannorm,position,scale]=mlGradient(img, nbin)
```

```
img(isinf(img))=NaN;
width = size(img, 2)
height = size(img, 1)
binsize = floor(width/nbin);
imgmean = zeros(nbin, 1);
imgstd = zeros(nbin,1);
for k = 1:nbin
if k ~= nbin
imgmean(k)=nanmean(img(:,(k-1)*binsize+1:k*binsize),'all');
imgstd(k)=std(double(img(:,(k-1)*binsize+1:k*binsize)),0,'all', 'omitnan');
else
imgmean(k)=nanmean((img(:,(k-1)*binsize+1:width)),'all');
imgstd(k)=std(double(img(:,(k-1)*binsize+1:width)),0,'all', 'omitnan');
end
end
scale = [0:1/nbin:1]';
position = [0+(1/(nbin*2)):1/nbin:1-(1/(nbin*2))]';
imgmeannorm = imgmean/imgmean(nbin);
end
```

## Sample size and significance test

Each experiment includes a minimum of three to five biological replicates. A biological replicate is equal one embryo whereas technical replicates represent separate experiments (e.g. staining reactions or timelapses). In *Figure 6*, the number of biological and technical replicates are equal. In other figures, we provide the number of biological replicates which derive from at least two technical replicates. In our experience, this sample size combined with detailed quantitative analysis of each sample will reveal any significant differences between experiment samples and controls. Samples were excluded from analysis if the signal to noise in the image data were too low to perform a quantitative analysis. To establish if there is any significant difference between samples in a metric, we used a two-sample T-test provided by the Matlab function' ttest2' or by standard 'ttest' function in Excel. Significance was defined as: $*p<0.05$, $**p<0.005$, and $***p<0.0005$.

## ImageJ macro for fitting circle

```
requires("1.43k");
if (nImages == 0)
exit("No image is open.");
showMessageWithCancel("Circle Fit Macro.", "This macro will reset your ROI Man-
ager.\nDo you
want to proceed?");
roiManager("reset");
}
setTool("multipoint");
waitForUser("Circle Fit Macro.", "Multipoint tool selected.\nSelect points in
your image, then click OK.");
roiManager("Add"); roiManager("Select", 0); roiManager("Rename", "points");
getSelectionCoordinates(x, y);
n = x.length;
if(n < 3)
exit("At least 3 points are required to fit a circle.");
sumx = 0; sumy = 0;
for(i = 0;i < n;i++) {
sumx = sumx + x[i];
```

```
sumy = sumy + y[i];
}
meanx = sumx/n;
meany = sumy/n;
X = newArray(n); Y = newArray(n);
Mxx = 0; Myy = 0; Mxy = 0; Mxz = 0; Myz = 0; Mzz = 0;
for(i = 0;i < n;i++) {
X[i]=x[i] - meanx;
Y[i]=y[i] - meany;
Zi = X[i]*X[i] + Y[i]*Y[i];
Mxy = Mxy + X[i]*Y[i];
Mxx = Mxx + X[i]*X[i];
Myy = Myy + Y[i]*Y[i];
Mxz = Mxz + X[i]*Zi;
Myz = Myz + Y[i]*Zi;
Mzz = Mzz + Zi*Zi;
}
Mxx = Mxx/n;
Myy = Myy/n;
Mxy = Mxy/n;
Mxz = Mxz/n;
Myz = Myz/n;
Mzz = Mzz/n;
Mz = Mxx + Myy;
Cov_xy = Mxx*Myy - Mxy*Mxy;
Mxz2 = Mxz*Mxz;
Myz2 = Myz*Myz;
A2 = 4*Cov_xy - 3*Mz*Mz - Mzz;
A1 = Mzz*Mz + 4*Cov_xy*Mz - Mxz2 - Myz2 - Mz*Mz*Mz;
A0 = Mxz2*Myy + Myz2*Mxx - Mzz*Cov_xy - 2*Mxz*Myz*Mxy + Mz*Mz*Cov_xy;
A22 = A2 + A2;
epsilon = 1e-12;
ynew = 1e+20;
IterMax = 20;
xnew = 0;
//Newton's method starting at x = 0for(iter = 1;i <= IterMax;i++) {
yold = ynew;
ynew = A0 + xnew*(A1 + xnew*(A2 + 4.*xnew*xnew));
if (abs(ynew)>abs(yold)) {
print('Newton-Pratt goes wrong direction: |ynew| > |yold|");
xnew = 0;
i = IterMax+1;
}
else {
Dy = A1 + xnew*(A22 + 16*xnew*xnew);
xold = xnew;
xnew = xold ynew/Dy;
if (abs((xnew-xold)/xnew)<epsilon)
i = IterMax+1;
else {
if (iter >= IterMax) {
print('Newton-Pratt will not converge');
xnew = 0;
}
if (xnew <0) {
//print('Newton-Pratt negative root: x = "+xnew);
```

```
xnew = 0;
}
}
}
}
DET = xnew*xnew - xnew*Mz + Cov_xy;
CenterX = (Mxz*(Myy-xnew)-Myz*Mxy)/(2*DET);
CenterY = (Myz*(Mxx-xnew)-Mxz*Mxy)/(2*DET);
radius = sqrt(CenterX*CenterX + CenterY*CenterY + Mz + 2*xnew);
if(isNaN(radius))
exit("Points selected are collinear.");
CenterX = CenterX + meanx;
CenterY = CenterY + meany;
print("Radius = "+d2s(radius,2));
print("Center coordinates: ("+d2s(CenterX,2)+", "+d2s(CenterY,2)+")");
print("(all units in pixel)");
makeOval(CenterX-radius, CenterY-radius, 2*radius, 2*radius);
roiManager("Add");  roiManager("Select",  1);  roiManager("Rename",  "circle
fit");
setTool("point");
makePoint(CenterX, CenterY);
roiManager("Add"); roiManager("Select", 2); roiManager("Rename", "center");
roiManager("Show All without labels");
```

## Acknowledgements

We thank Madhusudhan Venkadesan for insightful discussions and members of the Holley lab for critical comments on the manuscript. Research support from the NIH initially provided by R01GM107385 and later by R01HD092361 to SAH.

## Additional information

### Funding

| Funder | Grant reference number | Author |
|---|---|---|
| National Institute of General Medical Sciences | R01GM107385 | Scott Holley |
| Eunice Kennedy Shriver National Institute of Child Health and Human Development | R01HD092361 | Scott Holley |

The funders had no role in study design, data collection and interpretation, or the decision to submit the work for publication.

### Author contributions

Emilie Guillon, Conceptualization, Data curation, Formal analysis, Validation, Investigation, Visualization, Methodology; Dipjyoti Das, Conceptualization, Data curation, Software, Formal analysis, Methodology; Dörthe Jülich, Data curation, Formal analysis, Supervision, Investigation, Methodology; Abdel-Rahman Hassan, Formal analysis; Hannah Geller, Investigation; Scott Holley, Conceptualization, Formal analysis, Supervision, Funding acquisition, Project administration

### Author ORCIDs

Emilie Guillon https://orcid.org/0000-0002-6731-6689
Scott Holley https://orcid.org/0000-0001-5299-1174

### Ethics

Animal experimentation: Zebrafish were raised according to standard protocols (Nüsslein-Volhard and Dahm, 2002) and approved by the Yale Institutional Animal Care and Use Committee.

### Decision letter and Author response

Decision letter https://doi.org/10.7554/eLife.48964.sa1
Author response https://doi.org/10.7554/eLife.48964.sa2

## Additional files

### Supplementary files

• Transparent reporting form

### Data availability

All data generated or analyzed are included in the manuscript and supporting files.

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
