## [Decision Letter]

**Acceptance summary:**

This paper elegantly combines quantitative experimental measurements with an interesting lap-joint model to describe the formation of the spinal column in zebrafish. These insights will deepen our understanding of how complex tissue shape emerges during development.

**Decision letter after peer review:**

Thank you for submitting your article "Fibronectin is a smart adhesive that both influences and responds to the mechanics of early spinal column development" for consideration by *eLife*. Your article has been reviewed by two peer reviewers, including Timothy E Saunders as the Reviewing Editor and Reviewer #1, and the evaluation has been overseen by Didier Stainier as the Senior Editor. The other reviewer has opted to remain anonymous.

The reviewers have discussed the reviews with one another and the Reviewing Editor has drafted this decision to help you prepare a revised submission.

Summary:

In this work, Guillon et al., use concepts from engineering to deepen our understanding of how the early spinal column develops. They identify Fibronectin as a key player in determining how mechanical inputs are integrated during spinal column morphogenesis. This work further adds to the realization that inter-tissue interactions are essential in tissue morphogenesis. A particular stand-out result is the similarity in behavior of materials at macro- and micro- scopic scales.

The reviewers agree that the paper contains interesting new insights into how the early spinal cord develops in zebrafish and is potentially suitable for *eLife*. However, there are a few issues that need clarification, which are summarized here (see reports for more in-depth detail):

Essential revisions:

1) The model needs to be developed more clearly, in particular see point 4 of reviewer 1.

2) We do not see the need for additional experiments except better exploration of the link between Fibronectin and tissue tension (point 5 of reviewer 1). This is a key input of the model and needs better validation. Laser ablations are suggested, but not the only means of performing such a test.

3) Presentation of figure images can be improved, and, in particular, the results of Figure 6 need to be made more accessible.

4) The Discussion can be improved to bring out the key results of the paper in a more accessible manner.

Reviewer #1:

In general, this paper uses a number of different conceptual ideas to understand how the spinal column develops. It will likely appeal to developmental biologists interested in organogenesis, bioengineers, and physicists. Therefore, it has potential to have broad impact across a range of fields.

The paper in its current form has a number of issues that should be addressed:

1) Most of the paper is based on the convergence of neural tube defects observed in different genotypes (Figure 1). This data is important but the images in the printed version are poor. I am aware that Fibronectin is not an easy staining, but the staining of the nuclei can be significantly improved. It would help if the authors add supplementary data using a marker for the neural tube to show the convergence defect of neural tube in different genotypes. This will help give confidence in the presented results.

2) In general, standard deviation should be used to show the variability in the data (e.g. Figure 1F). This is important to gauge the reproducibility of data, which is masked by SEM.

3) In the Introduction, the concepts of lap-joints and spew filet need much clearer definitions. The current version is too brief.

4) Related to the model. a) Figure 2A: Is there a contribution of the skin to reach the steady tissue shapes?

b) The PSM and notochord are attached by a set of springs (red springs, which model inter-tissue adhesion via cell-Fibronectin interactions) to a rigid yolk surface (black line), In Figure 1 there is no Fibronectin between this tissue and the yolk, so which molecule links the PSM and notochord to the yolk?

c) Is the model able to reproduce the normal morphology of the neural tube along the PSM (anterior narrower than posterior)?

d) Is the model able to reproduce the morphology defects of the convergence of neural tube in the mutants?

e) It is not clear how viscosity has been incorporated within the model. If it has been excluded, better justification for this is needed.

f) Though "different parameters" are mentioned, more specificity is required. Was a parameter screen performed, or was more targeted parameter testing performed? This detail is important and should be in main text.

g) I appreciate that interpreting experimental results into parameter values is challenging (e.g. Figure 2D). It would be better to present these values as a range, and present the model results across a range of values.

5) The model predicts that there is tension at the tissue interface (gradient voltage PSM|NT interface, homogeneous voltage PSM|E interface). This should be directly tested by, for example, laser ablation experiments. Just inferring force from Fibronectin density is not sufficient.

6) The in vivo vs. in silico comparison in Figure 2E-F is not clear and can improved to aid in interpretation.

7) The Authors show that reductions of cell-cell adhesion eliminates the medial lateral gradients of Fibronectin matrix and F-actin in *cdh2*^-/-^; *MZ itgα5*^-/-^ mutant embryos (Figure 4). What happens to the gradients of Fibronectin matrix and F-actin in *cdh2*^-/-^; *MZ itgα5*^-/-^ double mutants embryos? Do they predict that the gradients are rescued?

8) Related to Figure 6, what is the explanation for the variable degree of neural convergence exhibited in *cdh*-^/-^ and *MZ itgα*^-/-^ embryo mutants?

9) Related to the Discussion. The Authors note that Folate deficiency causes spina bifida and increases cell traction force in neuronal cell culture (Kin et al., 2018). Can this idea be extended, to ask whether folate increases traction force? (this is not a request for further experiments, simply a broader discussion and a chance to highlight possible future experiments).

10) The work focuses on somite stages 12-14. It would be good to discuss how general these results are for different stages of neural column development.

Overall, the experiments are solid. However, the theory appears a little underdeveloped, in particular with regard to the morphology of the tissues in both wildtype and mutant scenarios (e.g. can the model graphically show the morphological defects in the asymmetry of the convergence of the neural tube in the different mutants). The discussion of the model in the paper is rather brief and some of these issues may be resolved by a more clear introduction of the model and the key assumptions underlying it. The model is a key part of the work and should be improved before acceptance.

Reviewer #2:

In this manuscript, the authors investigate the effects of cell adhesion mutations on neural tube convergence. The convergence defect of *cdh2* null mutants was rescued in a double null mutant (*cdh2*^-/-^;*itgα5*^-/-^) that also knocks out the Fibronectin receptor. Knock out of either Fibronectin or integrin α5 caused premature convergence. Together these results demonstrate that Fibronectin matrix provides resistance against neural tube convergence. Computational modeling was used to predict the effects of changes in adhesion on neural tube and presomitic mesoderm (PSM) shapes, tissue interactions, and surface tensions. These predictions were then tested in vivo leading to a lap joint model for Fibronectin matrix as an adhesive at the PSM-neural tube interface. A photoconvertible Fibronectin was used to measure changes in the matrix and the results showed that Fibronectin matrix remodeling depends on convergence. Some *cdh2*^-/-^ mutants showed asymmetric convergence. Fibronectin matrix remodeling in those cases was lost on the contralateral side, and this effect could be induced in the computational model by using different parameters for left and right sites. Overall this manuscript shows that Fibronectin matrix determines the symmetric morphogenesis of the spinal cord.

This is a very interesting manuscript with novel insights supporting surprising results. The novelty arises from the combination of computational modeling with experimental data interpreted through an engineering perspective. This approach identified a lap joint adhesion mechanism mediated by the Fibronectin matrix. The surprising finding was the contralateral effect of convergence on matrix remodeling, which suggests that an important and unanticipated role of cell-Fibronectin adhesion is symmetrical morphogenesis.

This manuscript is dense with a somewhat complicated story, especially for readers who might not be well-versed in developmental biology. To help readers understand the significant findings and conclusions from this work, it is recommended that the authors add a new first paragraph to the Discussion. This paragraph would briefly state the 3 or 4 main conclusions from their work. They can then use the remainder of the Discussion to say more about these conclusions. In the current version, readers are left more or less to their own devices to suss out what the key findings/conclusions are.

[Editors' note: further revisions were suggested prior to acceptance, as described below.]

Thank you for resubmitting your work entitled "Fibronectin is a smart adhesive that both influences and responds to the mechanics of early spinal column development" for further consideration by *eLife*. Your revised article has been evaluated by Didier Stainier as the Senior Editor, a Reviewing Editor and one peer reviewer.

The manuscript has been improved but we ask you to deal with one final concern.

The authors can improve the Discussion of the paper by clearly stating the 3 or 4 key experimental conclusions from the results that lead to the lap joint model. Examples of experimental conclusions include the following but it is up to the authors to decide what is most relevant – integrin mutants exhibit a narrower neural tube implicating cell-Fibronectin matrix adhesion in constraining neural tube convergence; in vivo testing of predictions from computational modeling support a role for Fibronectin matrix as an adhesive at the PSM-neural tube interface; symmetry depends on the balance of forces arising from NT convergence and cell-matrix adhesion and is lost when cell-Fibronectin interactions are perturbed. The final conclusion could be made more accessible to a broad audience by elaborating a bit. For example, is it accurate to say that the results show Fibronectin matrix determines the symmetric morphogenesis of the spinal cord and suggest a lap joint model for Fibronectin matrix as an adhesive at the PSM-neural tube interface?

---

## [Author Response]

Essential revisions:1) The model needs to be developed more clearly, in particular see point 4 of reviewer 1.

We have doubled the length of this section in the main text, revised one of the main figures and provided a new supplementary figure. These additions are detailed in the response point 4 of reviewer 1 below.

2) We do not see the need for additional experiments except better exploration of the link between Fibronectin and tissue tension (point 5 of reviewer 1). This is a key input of the model and needs better validation. Laser ablations are suggested, but not the only means of performing such a test.

We now provide new data showing gradients of Myosin-II localization and levels of a tension-dependent epitope in Fibronectin. Details are included in the response to point 5 from reviewer 1 below.

3) Presentation of figure images can be improved, and, in particular, the results of Figure 6 need to be made more accessible.

For Figure 6, we have added a description of how the measurements were made to the main text. See also our response to point 2 of reviewer 2 below.

4) The Discussion can be improved to bring out the key results of the paper in a more accessible manner.

To address this point, we have added a new first paragraph to the Discussion. See also our response to point 1 of reviewer 2 below.

In general, we have tried to balance reviewer 1’s request for additional information and reviewer 2’s comment that the manuscript already told a complicated story. Thus, while some parts of the manuscript have been expanded significantly, we sought to address other points as succinctly as possible.

Reviewer #1:[…] The paper in its current form has a number of issues that should be addressed:1) Most of the paper is based on the convergence of neural tube defects observed in different genotypes (Figure 1). This data is important but the images in the printed version are poor. I am aware that Fibronectin is not an easy staining, but the staining of the nuclei can be significantly improved. It would help if the authors add supplementary data using a marker for the neural tube to show the convergence defect of neural tube in different genotypes. This will help give confidence in the presented results.

We agree with reviewer 1 that for sizing purposes in the main figure, the images are quite small and with limited resolution. However, we have been segmenting the tailbud using only nuclear localization for years in our analysis cell motion in the tailbud. Here, we are able to supplement the nuclear stain with either Fibronectin immunolocalization or phalloidin staining which makes it even easier to segment the tissues. We also note that when we segment a single slice, we are able to use information in other planes to clarify ambiguities. As an example, we now provide a video for wild type (now Video 1) showing with a higher resolution the smooth progression (in transverse sections every 1μm) of tissue shapes from posterior to anterior. This video reveals easily identifiable tissue boundaries as well as the convergence of the neural tube.

2) In general, standard deviation should be used to show the variability in the data (e.g. Figure 1F). This is important to gauge the reproducibility of data, which is masked by SEM.

We have replaced SEM with SD calculations in the revised manuscript.

3) In the Introduction, the concepts of lap-joints and spew filet need much clearer definitions. The current version is too brief.

We have changed the last paragraph of the Introduction to include the following sentences to provide a definition of a lap joint and spew filet:

Our model recapitulates features of an ‘adhesive lap joint’ which is commonly used in engineering and is comprised of partially overlapping components bound via an adhesive. Excess adhesive that can ooze from the edges of the overlapping domains is called an ‘adhesive spew’ which can be filleted or sculpted to strengthen the lap joint. Here, the Fibronectin matrix functions as the adhesive in the lap joint formed by the neural tube and left and right paraxial mesoderm.

4) Related to the model. a) Figure 2A: Is there a contribution of the skin to reach the steady tissue shapes?

The epidermis could potentially mechanically constrain both the neural tube (NT) and presomitic mesoderm (PSM). However, since both the NT and PSM are surrounded by the epidermis, we postulate that it alters the effective surface stiffness of both of them to a similar degree. In a previous study on the effects of abrogating cell-Fibronectin interaction on cell migration, we found no changes in epidermis cell motion in the tailbud (Dray et al., 2013). Here, we observe no obvious change in the cell shapes of the epidermis across different genotypes, again suggesting that the mechanical properties of the skin may not vary drastically from one genotype to another. In our model, we focus on the shape changes of the NT and PSM relative to each other. Hence, to make a simple model with minimal assumptions, we neglect the absolute contribution of the epidermis.

b) The PSM and notochord are attached by a set of springs (red springs, which model inter-tissue adhesion via cell-Fibronectin interactions) to a rigid yolk surface (black line), In Figure 1 there is no Fibronectin between this tissue and the yolk, so which molecule links the PSM and notochord to the yolk?

This is an interesting question. There are low levels of Fibronectin between the yolk and overlying tissues (see Author response image 1), but the tissues remain attached to the yolk in the Fibronectin double mutant. We detect no Laminin by immunohistochemistry along this interface at this stage of development. Thus, there are likely other proteins at this interface that mediate adhesion.

**Author response image 1. respfig1:** Dashed box represents the PSM/yolk interface.

c) Is the model able to reproduce the normal morphology of the neural tube along the PSM (anterior narrower than posterior)?

Our aim was to model the final steady-state shapes of the NT-PSM interface in a 2D transverse section, close to the last somite boundary (i.e. anterior end of PSM). To keep the model as simple as possible, we did not explicitly model convergent extension. Our simple computational model has three main parameters: the internal pressure within a tissue, the surface stiffness, and the inter-tissue adhesion. To effectively model the NT convergence, we would need to go beyond the simple 2D coarse-grained model and explicitly include cell-cell intercalation in 3D.

Nonetheless, using the current model, we can assume that the NT-PSM interface is locally at steady-state along the anterior-posterior axis, and the observed tissue shape change along the anterior-posterior axis may be caused by a progressive variation of one of the model parameters along that axis. The zebrafish PSM progressively solidifies from posterior to anterior via a *cadherin2*-dependent gradient of yield-stress along the anterior-posterior axis (Mongera et al., 2018). Since *cadherin2* promotes cohesion in a tissue and high cohesion correlates with high tissue surface tension (David et al., 2014; Manning et al., 2010), the PSM solidification can represented by a gradual increase of surface stiffness in our coarse-grained model. In a new set of simulations, we found that the length of the medial to lateral interface between the PSM and NT (L PSM|NT) steadily decreases with increasing surface stiffness when other parameters are fixed (see Figure 2—figure supplement 1A). We observe a similar decrease in L PSM|NT from posterior to anterior in vivo. Thus, our 2D model can explain the in vivo tissue narrowing with the simple assumption of local steady-state and variation of surface stiffness from posterior to anterior.

An abbreviated version of this explanation is now included in the model section of the main text.

d) Is the model able to reproduce the morphology defects of the convergence of neural tube in the mutants?

In the previous response (4c), we noted the limitations of our model. Our model does not include convergent extension and does reproduce the morphological defects in NT convergence.

e) It is not clear how viscosity has been incorporated within the model. If it has been excluded, better justification for this is needed.

Viscosity is included in our model as we considered over-damped dynamics of the surface points that ultimately reaches a steady state (see the equation of motion in Materials and methods: Simulation methods). The viscosity, however, would only affect the timescale of reaching the steady state. Since we are primarily interested in relative tissue shapes in the steady-state, the absolute value of the viscosity is not important. In simulations, we found that the relative tissue shapes are unaltered with over a 10-fold variation of the viscous coefficient (*c*=5 to 50). Thus, we fixed the value of the viscous coefficient to *c*=10.

f) Though "different parameters" are mentioned, more specificity is required. Was a parameter screen performed, or was more targeted parameter testing performed? This detail is important and should be in main text.

The main parameters of our model are (i) internal pressure, (ii) surface stiffness, and (iii) inter-tissue adhesion. None of these parameters can be measured directly. Hence, we had to indirectly infer the parameter values from in vivo tissue shapes of the NT and PSM relative to each other, and from the relative shapes of their interface across the genotypes.

First, the internal tissue pressure should depend on the net fluid content inside a tissue, and there is no obvious reason to vary this quantity across genotypes. Hence, we fixed the internal pressure (P=5) depending on an earlier model of soft grains (Åström and Karttunen, 2006) in such a way that the simulated shapes qualitatively resemble in vivo tissue shapes in 2D transverse sections.

The remaining two parameters were then systematically varied in simulations to check their influence on the shape of the interface between NT and PSM. We found that increasing surface stiffness decreases the interfacial length, while increasing inter-tissue adhesion increases interfacial length (Figure 2—figure supplement 1A and B). As noted in the answer to point 4c, a posterior to anterior gradient in surface stiffness can be thought to mimic the solidification of the PSM, and thus the simulation result (Figure 2—figure supplement 1A) parallels with the in vivo decrease of interfacial length from posterior to anterior (Figure 2—figure supplement 1C). On the other hand, variation in either surface stiffness or adhesion stiffness in simulations did not show any systematic change in the interfacial angle (Figure 2—figure supplement 1D and E). This also parallels our in vivo observation that the interfacial angle does not vary along the anterior-posterior axis (Figure 2—figure supplement 1F).

Given the above correspondence between in vivo and in silico observations, we next assigned reasonable values of surface stiffness (K_S_) and inter-tissue adhesion stiffness (K_adh_) that reproduce morphometrics of wild-type embryos. We note that in vivo the value of the interfacial length (L PSM|NT) decreases to 0.6 of the maximum value at the anterior end of PSM relative to the posterior end (Figure 2—figure supplement 1C). We also found in silico that L PSM|NT roughly falls to 0.6 of the maximum value at K_S_ ≈ 100 for a fixed K_adh_ (Figure 2—figure supplement 1A). Since we are interested in steady-state shapes near the anterior PSM, we may take this value to represent the wild type. Next, we consider the value of K_adh_. In simulations, we varied K_adh_ for different fixed values of K_S_, and measured the medial-lateral length of NT relative to the interfacial length between NT and PSM (Figure 2—figure supplement 1G). In vivo, this ratio (ML length of NT/ L PSM|NT) is about 2 on average for the anterior 140 μm of PSM. In silico, we found that irrespective of the K_S_ value, this ratio saturates around a value of 2.7 in the range of K_adh_=8 to12 (marked in red in Figure 2—figure supplement 1G). Based on the above analysis, we then represented the wild-type embryos by a pair of values: (K_S_,K_adh_)=(100,10).

After fixing the wild-type parameters, we then chose the parameter values corresponding to *cdh2* mutants and *MZ itgα5* mutants by matching in silico the in vivo length of PSM|NT interface in these mutants relative to its wild-type values (Figure 2E). For *cdh2* mutants, the mean length of PSM|NT is around 1.5 times higher than wild type. To reproduce the experimentally observed increase of L PSM|NT relative to wild-type, we assigned reasonable parameter values to *cdh2* mutants depending on biological expectations. First, we lowered the surface stiffness relative to wild type, since low cell-cell adhesion is known to reduce tissue surface tension (David et al., 2014, Manning et al., 2010). Second, Cadherin 2 was shown to inhibit Integrin α5 activation and Fibronectin matrix assembly in the PSM (Jülich et al., 2015, McMillen et al., 2016). Therefore, we may assign a higher adhesion stiffness value to *cdh2* mutants compared to wild type. After systematic exploration of the parameter space in simulations, we found a pair of parameter values, (KS,Kadh)=(55,12), that reproduce the in vivo increase of L PSM|NT relative to wild-type (1.55 times).

Following the same procedure for *MZ itgα5* mutants, the in vivo data indicate that *MZ itgα5* mutants exhibit a mean PSM|NT interface length 0.77 times that of the wild-type value. Since cell-matrix interaction is reduced in *MZ itgα5* mutants, it is logical to assume a lower inter-tissue adhesion for this genotype relative to wild type, but the surface stiffness was kept same as wild type. We found that a pair of parameter values, (KS,Kadh)=(100,6.5), reproduce the experimentally observed decrease of L PSM|NT relative to wild-type (0.78 times).

Using the parameter values corresponding to *cdh2* mutants and *MZ itgα5* mutants, we can then predict the interfacial length for *cdh2; MZ itgα5* mutants simply by combining these parameter values. The double mutants are represented using the value of inter-tissue adhesion in *MZ itgα5* mutants, and the same value of surface stiffness as *cdh2* mutants (Figure 2D). Hence, *cdh2; MZ itgα5* mutants are represented by the values: (KS,Kadh)=(55,6.5). These parameter choices for *cdh2* mutants and *MZ itgα5* mutants accurately predicted the in vivo interfacial length of *cdh2; MZ itgα5* mutants, (Figure 2E). Importantly, though we assigned the parameter values depending on the relative interfacial lengths of *cdh2* mutants and *MZ itgα5* mutants, these parameter choices also reproduced the experimentally observed trends in the variation of interfacial angle across genotypes (Figure 2F).

This explanation is now included in the model section of the main text.

g) I appreciate that interpreting experimental results into parameter values is challenging (e.g. Figure 2D). It would be better to present these values as a range, and present the model results across a range of values.

In the previous response, we described how parameter values were assigned to different genotypes as in Figure 2D. In addition, we now provide systematic variation of surface stiffness and inter-tissue adhesion stiffness in simulations and show that these two parameters have opposing effects on the interfacial length (Figure 2—figure supplement 1). For the sake of simplicity, Figure 2D still only notes the specific values used for the final set of simulations that predict the double mutant phenotype and variation in interfacial angle across genotypes.

5) The model predicts that there is tension at the tissue interface (gradient voltage PSM|NT interface, homogeneous voltage PSM|E interface). This should be directly tested by, for example, laser ablation experiments. Just inferring force from Fibronectin density is not sufficient.

Although it is well known from cell culture experiments that Fibronectin density increases with F-actin levels, Myosin levels and cell traction forces, we agree that it would be good to have additional data to substantiate the prediction from our *in silico* model that there is a gradient of tension at the PSM|NT interface. We have now performed, two additional experiments to substantiate our conclusion. First, we quantified the spatial distribution of Myosin II in live embryos expressing myl12.1-EGFP (Araya et al., 2019; Behrndt et al., 2012) and found an increasing medial to lateral gradient of Myosin II, which is in accordance with F-actin density and Fibronectin matrix density (Figure 3—figure supplement 1A-C). Second, we probed the tension within the Fibronectin matrix itself by using the H5 antibody which recognizes an epitope that is exposed when human Fibronectin is under tension (Cao et al., 2017). For this experiment, we generated zebrafish embryos that express a chimeric zebrafish/human Fibronectin (FN1a-mKIKGR13.2-hsFNIII10-11 mRNA) (Figure 3—figure supplement 1D-M). We found that the PSM|NT interface exhibits an increasing medial-lateral gradient of H5 signal suggesting that the aggregate stress applied to the Fibronectin matrix is higher on the lateral side of the PSM|NT interface.

These new data are included in a new Figure 3—figure supplement 1.

6) The in vivo vs. in silico comparison in Figure 2E-F is not clear and can improved to aid in interpretation.

As part of our response to point 4f, the content of Figure 2E-F have been revised, and we think that the presentation will aid interpretation of the data.

7) The Authors show that reductions of cell-cell adhesion eliminates the medial lateral gradients of Fibronectin matrix and F-actin in cdh2^-/-^; MZ itgα5^-/-^ mutant embryos (Figure 4). What happens to the gradients of Fibronectin matrix and F-actin in cdh2^-/-^; MZ itgα5^-/-^ double mutants embryos? Do they predict that the gradients are rescued?

Figure 4 describes the phenotypes of the *cdh2* and *MZ itgα5* single mutants. The reviewer’s question, as we understand it, is what happened to the gradients in the double mutant?

The convoluted morphology and irregular ECM deposition in the double mutant mean that we cannot apply the same image segmentation protocol to the double mutants as for the other genotypes. Thus, we have not completed the parallel analysis of fixed double mutant embryos. However, based on the results of photoconversion experiments in single *MZ itgα5*^-/-^ mutants showing that the medio-lateral remodeling of the matrix depends on tissue attachment, one could think that any rescue F-actin and Fibronectin gradients in *cdh2*^-/-^; *MZ itgα5*^-/-^ double mutants will probably locally depends on the degree of tissue attachment. If tissues are still “relatively” attached, we can probably expect “normal” gradients of tension and Fibronectin matrix as suggested by the medio-lateral remodeling of the Fibronectin matrix in our photoconversion experiments in *cdh2*^-/-^; *MZ itgα5*^-/-^ double mutants.

8) Related to Figure 6, what is the explanation for the variable degree of neural convergence exhibited in cdh^-/-^ and MZ itgα^-/-^ embryo mutants?

We observed a variable neural tube convergence defect in single *cdh2-/-* mutants and in the *cdh2-/-; MZ itgα5-/-* double mutants. Figure 6 shows that MZ *itgα5-/-* have variable tissue detachment phenotype. We interpret the reviewer’s question with regards to the variability of the neural convergence phenotype in the double mutant.

If the tissue detachment phenotype of single *MZ itgα5 -/-* mutants were complete, then one might expect that the *cdh2-/-; MZ itgα5-/-* double mutants would have uniformly converging neural tubes. Thus, some of the variability of the neural tube convergence defect is likely due to variable tissue detachment and the resulting variable shear stress between the neural tube and paraxial mesoderm. The variable tissue detachment phenotype due to loss of *itgα5* is compounded by the intrinsic variability of the neural tube convergence defect in *cdh2* mutants. At present, it is not clear why the neural convergence phenotype in the *cdh2-/-* single mutant is so variable, but that is an interesting question.

9) Related to the Discussion. The Authors note that Folate deficiency causes spina bifida and increases cell traction force in neuronal cell culture (Kin et al., 2018). Can this idea be extended, to ask whether folate increases traction force? (this is not a request for further experiments, simply a broader discussion and a chance to highlight possible future experiments).

We have added a sentence to this point in the Discussion.

10) The work focuses on somite stages 12-14. It would be good to discuss how general these results are for different stages of neural column development.

We added four sentences to the third paragraph of the Discussion to elaborate on this topic.

Reviewer #2:[…] This manuscript is dense with a somewhat complicated story, especially for readers who might not be well-versed in developmental biology. To help readers understand the significant findings and conclusions from this work, it is recommended that the authors add a new first paragraph to the Discussion. This paragraph would briefly state the 3 or 4 main conclusions from their work. They can then use the remainder of the Discussion to say more about these conclusions. In the current version, readers are left more or less to their own devices to suss out what the key findings/conclusions are.

We modified the Discussion section to include the following paragraph to briefly state our main conclusions and help the readers:

“Here we find that inter-tissue adhesion, mediated by a Fibronectin matrix, mechanically couples the neural tube and adjacent mesoderm. […] Thus, Fibronectin functions as a ‘smart adhesive’ that continually remodels to where it is most needed.”

[Editors' note: further revisions were suggested prior to acceptance, as described below.]The manuscript has been improved but we ask you to deal with one final concern.The authors can improve the Discussion of the paper by clearly stating the 3 or 4 key experimental conclusions from the results that lead to the lap joint model. Examples of experimental conclusions include the following but it is up to the authors to decide what is most relevant – integrin mutants exhibit a narrower neural tube implicating cell-Fibronectin matrix adhesion in constraining neural tube convergence; in vivo testing of predictions from computational modeling support a role for Fibronectin matrix as an adhesive at the PSM-neural tube interface; symmetry depends on the balance of forces arising from NT convergence and cell-matrix adhesion and is lost when cell-Fibronectin interactions are perturbed. The final conclusion could be made more accessible to a broad audience by elaborating a bit. For example, is it accurate to say that the results show Fibronectin matrix determines the symmetric morphogenesis of the spinal cord and suggest a lap joint model for Fibronectin matrix as an adhesive at the PSM-neural tube interface?

The Editors made two suggestions regarding the Discussion section of the prior submission. The first suggestion was to provide an explication of the evidence supporting the lap joint model. We now provide this information in the second paragraph of the Discussion. The second suggestion was to restate the major conclusions in more general terms, which we now do in a new paragraph that concludes the Discussion.

We used ‘track changes’ to highlight the revisions in the Discussion section. The first three paragraphs have been substantially edited and the last paragraph is new. The second to last and third to last paragraphs were moved so that they follow the paragraph on spina bifida, rather than preceding the spina bifida paragraph as in the prior version of the manuscript. We think that this new organization of the Discussion helps address the Editorial comments.